# The retrotrapezoid nucleus neurons expressing *Atoh1* and *Phox2b* are essential for the respiratory response to $CO_2$

Pierre-Louis Ruffault[1], Fabien D'Autréaux[2], John A Hayes[1], Marc Nomaksteinsky[2], Sandra Autran[1], Tomoyuki Fujiyama[3,4], Mikio Hoshino[3], Martin Hägglund[5], Ole Kiehn[5], Jean-François Brunet[2]*, Gilles Fortin[1]*, Christo Goridis[2]

[1]Université Paris-Saclay, Université Paris-Sud, CNRS, UMR 9197, Institut des Neurosciences Paris-Saclay, Gif-sur-Yvette, France; [2]Institut de Biologie de l'École Normale Supérieure, Inserm U1024, and CNRS UMR 8197, Paris, France; [3]Department of Biochemistry and Cellular Biology, National Institute of Neuroscience, National Center of Neurology and Psychiatry, Tokyo, Japan; [4]International Institute for Integrative Sleep Medicine, University of Tsukuba, Tsukuba, Japan; [5]Department of Neuroscience, Karolinska Institutet, Stockholm, Sweden

**Abstract** Maintaining constant $CO_2$ and $H^+$ concentrations in the arterial blood is critical for life. The principal mechanism through which this is achieved in mammals is the respiratory chemoreflex whose circuitry is still elusive. A candidate element of this circuitry is the retrotrapezoid nucleus (RTN), a collection of neurons at the ventral medullary surface that are activated by increased $CO_2$ or low pH and project to the respiratory rhythm generator. Here, we use intersectional genetic strategies to lesion the RTN neurons defined by *Atoh1* and *Phox2b* expression and to block or activate their synaptic output. Photostimulation of these neurons entrains the respiratory rhythm. Conversely, abrogating expression of Atoh1 or Phox2b or glutamatergic transmission in these cells curtails the phrenic nerve response to low pH in embryonic preparations and abolishes the respiratory chemoreflex in behaving animals. Thus, the RTN neurons expressing *Atoh1* and *Phox2b* are a necessary component of the chemoreflex circuitry.

*For correspondence: jfbrunet@ biologie.ens.fr (J-FB); fortin@inaf. cnrs-gif.fr (GF)

**Reviewing editor**: Marlene Bartos, Albert-Ludwigs-Universität Freiburg, Germany

## Introduction

In mammals, breathing is the prime homeostatic process that regulates $CO_2$ partial pressure ($PCO_2$) in the blood. The respiratory chemoreflex modulates ventilation in response to deviations in arterial or brain $PCO_2$, mainly through the detection of changes in pH (*Feldman et al., 2003*; *Guyenet et al., 2010*). These changes are sensed by central chemosensors located in the brainstem and by the carotid bodies in the periphery, but most of the $CO_2$ chemosensory drive to breathe is thought to arise centrally (*Smith et al., 2006*; *Duffin, 2010*). The nature of the central chemosensors and of the circuits that mediate the ventilatory response to $CO_2$ is still controversial (*Guyenet et al., 2010*; *Huckstepp and Dale, 2011*; *Nattie, 2011*; *Guyenet et al., 2013*). Two sites have been proposed to play a dominant role in the central $CO_2$ response: serotonergic neurons in the medulla (*Ray et al., 2011*; *Teran et al., 2014*) and the retrotrapezoid nucleus (RTN), a loose collection of neurons located ventral and immediately caudal to the facial nucleus (*Goridis et al., 2010*; *Guyenet et al., 2013*). RTN neurons—defined by their location, by the expression of Phox2b and of the vesicular glutamate transporter-2 Vglut2 (also known as Slc17a6) and by the absence of markers for the nearby motor and

**eLife digest** An adult at rest will typically breathe in and out up to 20 times per minute, inhaling oxygen and exhaling carbon dioxide in a process that, for the most part, occurs automatically. While we can choose to override this process and exert voluntary control over our breathing, we cannot suppress it indefinitely. Attempting to do so will ultimately trigger a reflex that forces us to start breathing again.

This reflex is mostly a response to the rise of carbon dioxide ($CO_2$) in the blood, which lowers the pH of the blood. This rise in $CO_2$ is toxic and triggers an increase in breathing so that the excess $CO_2$ is exhaled. The majority of the sensors that detect $CO_2$ are in the brainstem, which is at the junction of the brain and the spinal cord. However, the precise location of these sensors is not clear. Ruffault et al. now argue that the sensors are in a region called the 'retrotrapezoid nucleus', and that they can be identified by the presence of two proteins, Atoh1 and Phox2b.

In the brains of foetal mice, Ruffault et al. recorded cells in the retrotrapezoid nucleus and found that they fired in a rhythmic pattern, as would be expected for cells that control breathing. Moreover, the firing rate of these cells increased when the pH was lowered.

Ruffault et al. then created genetically modified mice with mutations in genes for Atoh1 or Phox2b. The retrotrapezoid nucleus was either absent or abnormal in these mutant mice. Moreover, new-born pups with these mutations were not able to increase their breathing when the level of $CO_2$ in their blood rose.

These results shed light on the respiratory distress experienced by patients with a rare disorder called congenital central hypoventilation syndrome (CCHS) that is caused by mutations in Phox2b. More commonly, unstable or irregular breathing is seen in human infants that are born prematurely, and sometimes in infants born at full term. In the light of the new findings by Ruffault et al., it is possible that abnormal development or immaturity of the retrotrapezoid nucleus is the cause.

catecholaminergic neurons—are vigorously activated by increases in $CO_2$ (*Mulkey et al., 2004*; *Onimaru et al., 2012b*; *Wang et al., 2013*). Neurons with the molecular characteristics of RTN neurons are already present ventral of the facial nucleus in the late embryo. The embryonic RTN neurons, previously termed embryonic parafacial cells, were found to have endogenous bursting properties and to couple with and increase the frequency of the respiratory rhythm generating (*Feldman et al., 2013*) pre-Bötzinger complex (pre-BötC) (*Dubreuil et al., 2009*; *Thoby-Brisson et al., 2009*).

Impairment of RTN development or function correlates with a blunted response to hypercapnia or acidification (*Nattie and Li, 2002*; *Dubreuil et al., 2009*; *Marina et al., 2010*; *Ramanantsoa et al., 2011*; *Takakura et al., 2013*, *2014*), and mouse models of congenital central hypoventilation syndrome (CCHS), which phenocopy the blunted or absent chemoreflex of the patients (*Weese-Mayer et al., 2010*; *Rossor et al., 2014*), have an atrophic RTN (*Dubreuil et al., 2008*; *Ramanantsoa et al., 2011*). However, the part played by the RTN in the respiratory chemoreflex is still highly debated, and the fraction of the $CO_2$ response that is relayed through RTN neurons is undetermined (*Guyenet, 2014*). In a recent study, mutants with a damaged RTN displayed a normal CO2/pH sensitivity at embryonic stages in vitro (*Huang et al., 2012*).

To resolve the issue of the requirement for RTN neurons in the respiratory chemoreflex, we selectively impaired RTN development or function using intersectional genetic strategies (*Dymecki et al., 2010*) and assessed the functional consequences at pre- and post-natal stages. To target the Phox2b-expressing RTN neurons with utmost specificity, we exploited the fact that the majority of them belongs to the small set of neurons that co-express the Phox2b and Atoh1 transcription factors in the embryo (*Dubreuil et al., 2009*; *Rose et al., 2009b*). We found that invalidation of *Atoh1* in *Phox2b*[+] cells, of *Phox2b* in *Atoh1*[+] cells or of glutamatergic transmission in *Phox2b*[+]/*Atoh1*[+] cells curtailed the response to acidification in embryonic brainstem preparations and eliminated the $CO_2$ response in newborn pups. Optogenetic activation of these cells entrained ongoing respiratory rhythm. Together our findings imply that the RTN neurons that express or have expressed *Phox2b* and *Atoh1* are essential for the activation of breathing by increased $CO_2$ or low pH, and that other contributors to the chemoreflex must act via the RTN or in partnership with it.

## Results

### Photoactivation of the embryonic RTN entrains the respiratory motor output

We first verified the ability of embryonic RTN neurons to entrain the respiratory-like motor output using channelrhodopsin-based optogenetics. The Phox2b+ RTN neurons are glutamatergic (*Bochorishvili et al., 2012*) and express the glutamate transporter *Vglut2* already at embryonic stages (*Dubreuil et al., 2009*). We could thus use expression of the channelrhodopsin-2-YFP (ChR2-YFP) fusion protein driven by the *Vglut2* promoter in *Vglut2::ChR2-YFP* mice (*Hägglund et al., 2010*) to stimulate embryonic RTN neurons by light. In the transgenic embryos, cells co-expressing ChR2-YFP and Phox2b were concentrated at the medullary surface ventral to the ChR2-YFP-negative facial neurons, thus well accessible to light delivered from the ventral surface (*Figure 1A–C*). At embryonic-day 14.5 (E14.5), single light pulses (473 nm, 70 ms, 1–5 mW/mm$^2$) applied to the RTN region in brainstem preparations systematically evoked a burst of action potentials in ChR2-YFP expressing cells (n = 5) that resembled the spontaneous rhythmic bursts (*Figure 1D*). A latency of 192 ± 12 ms (n = 51 stimulations in three cells) was measured from the time of onset of the light stimulus to that of the first action potential of the burst response suggesting the requirement of a still unknown, slow obligatory process for burst initiation in the RTN. 1 day later, when the preBötC is coupled to the RTN oscillator and drives a respiratory-like motor outflow (*Thoby-Brisson et al., 2009*), single light pulses (473 nm, 150 ms, 1–5 mW/mm$^2$) delivered to the medullary surface triggered motor bursts in the C4 phrenic nerve roots (hereafter C4). When the light pulses were set to activate the RTN in a rhythmic manner at about twice the frequency of the ongoing endogenous rhythm, the C4 motor bursts could be entrained to the stimuli and followed the light-imposed rhythm (*Figure 1E*). C4 motor bursts could not be evoked when the preBötC excitability was depressed by the μ-opiate agonist D-Ala$^2$-N-Me-Phe$^4$-Glycol$^5$-enkephalin (DAMGO, 0.3 μM, n = 4 preparations) (*Mellen et al., 2003*) (*Figure 1F*) or its development impaired genetically in *Dbx1* null mutants (*Bouvier et al., 2010*) (n = 5 preparations) (*Figure 1G*) indicating that the motor outputs require an intact preBötC. These data suggest that malfunction of the RTN will result in lack of entrainment of the preBötC and thus of the motor output and in a slowed-down C4 activity.

### *Atoh1* is essential for CO$_2$/pH sensitivity

*Atoh1* null mutants (hereafter *Atoh1$^{-/-}$*) do not establish a proper respiratory rhythm and die at birth of respiratory failure (*Rose et al., 2009b*), but the development of the RTN and CO$_2$ chemosensitivity have not been examined in these mutants. In these and subsequent experiments, we used calcium imaging to monitor the rhythmic activity and response to acidification of embryonic brainstem preparations incubated in artificial cerebrospinal fluid (a-CSF) containing either Calcium Green-1AM or Fluo-8 AM as calcium indicators. Consistent with our previous observations (*Dubreuil et al., 2009*; *Thoby-Brisson et al., 2009*) calcium imaging of E14.5 *Atoh1$^{+/-}$* brainstem preparations shows rhythmically active RTN neurons that increase their bursting frequencies in response to acidification. These cells were absent in the *Atoh1$^{-/-}$* mutants and could not be revealed by low pH (*Figure 2A,B*). At E16.5, the mutants displayed a slowed-down respiratory-like rhythm in the C4 nerve roots (by an average of 56%, p < 0.001, n = 11 and 8 for control and mutants, respectively) and a complete lack of response to acidification (p = 0.1) (*Figure 2C,D*). We then monitored breathing parameters by plethysmography in E18.5 pups delivered by Caesarean section (*Figure 2E,F*). In the mutants, respiratory patterns ranged from hardly any breathing movements to slow rhythmic breathing that were completely unresponsive to hypercapnia (p = 0.96, n = 31 and 8 for control and mutants, respectively) (*Figure 2G–I* and *Table 1*). Together, the results show that *Atoh1* is essential for the formation of a functional RTN and for CO$_2$ chemical drive to breathe before and at birth.

The malformation of the RTN in the mutants may underpin the lack of CO$_2$ responsiveness. In *Atoh1$^{-/-}$* embryos, most RTN precursors, whether defined by expression of the *Atoh1* locus and Phox2b or by co-expression of Lbx1 and Phox2b, fail to migrate to the medullary surface and accumulate instead dorso-laterally of the facial nucleus (*Figure 3A–F,I*), consistent with previous findings (*Rose et al., 2009b*). In the mutants, 83 ± 1.5% of the *Atoh1*-expressing periVII cells were found at a dorsal location compared to 32 ± 4.1% in the controls (p < 0.001, n = 3). By contrast, the cells co-expressing the *Atoh1* locus and Phox2b that surround the trigeminal motor nucleus are present in the mutants at their normal location (p = 0.33, n = 3) (*Figure 3G,H,J*).

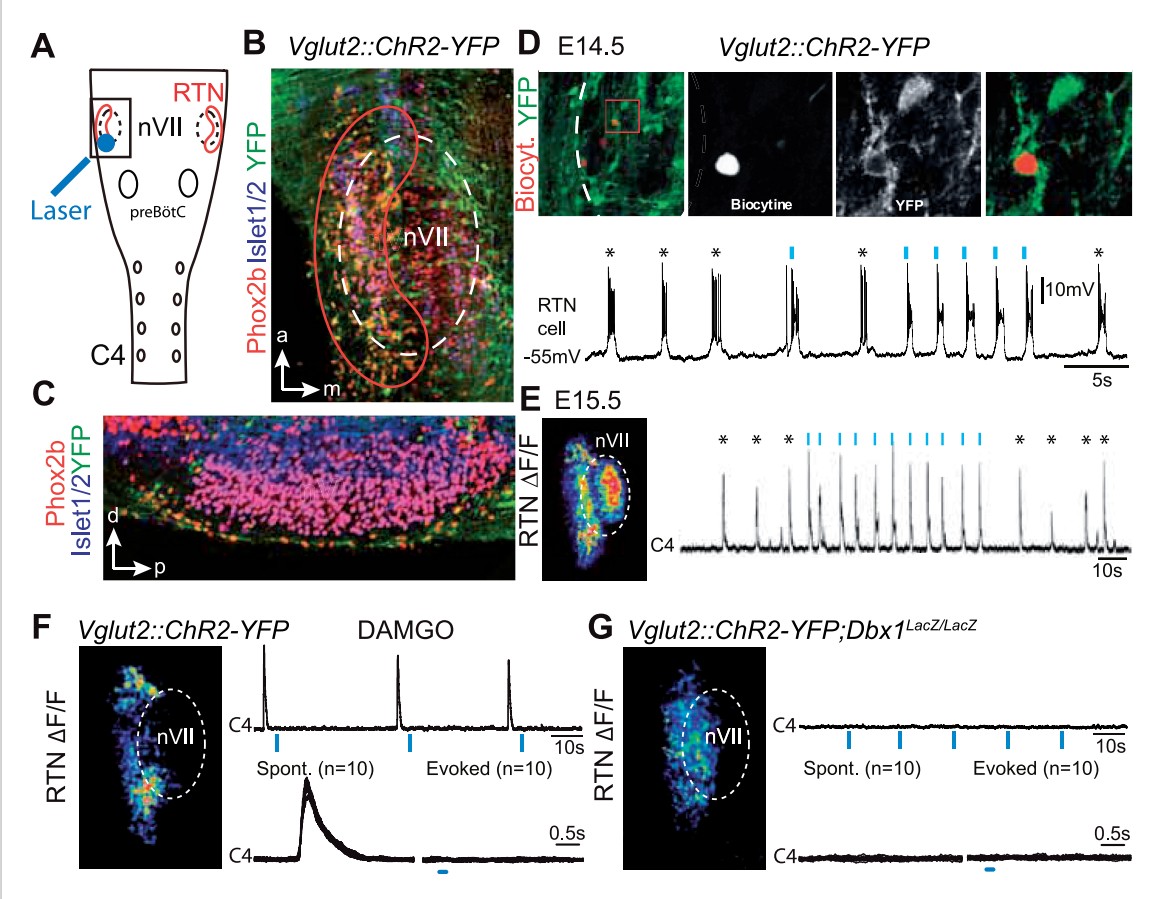

**Figure 1.** Effect of photostimulating Vglut2::Chr2-expressing embryonic retrotrapezoid nucleus (RTN) neurons on membrane potential and motor output. (**A**) Cartoon of a ventral view of the hindbrain showing the position of nVII (dotted line), RTN (red outline), the preBötC and the C4 phrenic nerve roots. (**B**) Ventral view corresponding to the boxed area in **A** (anterior at top, lateral on the left) of the RTN (red outline) and nVII (stippled outline) in a *Vglut2::ChR2-YFP* embryo, labeled for the indicated markers. (**C**) Sagittal section, labeled for the indicated markers, showing the Phox2b+/YFP+/Islet1,2− cells ventral to nVII and optimally accessible to light. (**D**) Above, ventral view of YFP expression in the RTN region of an E14.5 *Vglut2::ChR2-YFP* embryo showing the three recorded and biocytin-filled RTN cells (left panel). A high magnification of the one in the boxed area is provided in the right panels showing biocytin and YFP labeling separately and the overlay. Below, membrane potential trajectory of the RTN cell shown at high magnification, featuring spontaneously rhythmic (stars) and light-evoked (blue bars) bursts of action potentials. (**E**) Left panel, photostimulation of the RTN region in E15.5 brainstem-spinal cord preparations from a *Vglut2::ChR2-YFP* embryo evokes Calcium Green-1 AM (ΔF/F) changes in the RTN and adjacent nVII. Right panel, integrated C4 recording in such a preparation showing spontaneous C4 bursts of activity (stars) and bursts evoked by photostimulation (blue bars). Note that the light pulses entrain the ongoing endogenous rhythm. (**F**) In the presence of DAMGO that depresses the excitability of the preBötC, but not of the RTN, photostimulation of the RTN region in E15.5 preparations still evokes ΔF/F changes in the RTN but not in nVII (left panel). Right panel, above, integrated C4 activity showing systematic failures of photostimulation to evoke C4 responses. Below, set of 10 superimposed C4 activity traces. (**G**) Same experiment in *Vglut2::ChR2;Dbx1^{LacZ/LacZ}* mutants (n = 5) in which the preBötC but not the RTN is disrupted. The mutation completely abrogates light-evoked nVII and C4 responses.

## Designing a genetic strategy to target RTN neurons

In the brainstem, *Atoh1* is expressed in the rhombic lip progenitors and its early differentiating progeny and essential for their specification. A second, independent site of *Atoh1* expression is post-mitotic, in cells that co-express the Phox2b transcription factor and surround the trigeminal motor nucleus (hereafter periV neurons) and the facial nucleus (hereafter periVII neurons, which include a majority of the RTN neurons) (*Dubreuil et al., 2009*; *Rose et al., 2009b*). The lack of $CO_2$ responsiveness of $Atoh1^{-/-}$ embryos could thus be caused by malfunction of the RTN or some other *Atoh1*-dependent structure. Similarly, the *Phox2b* mutant backgrounds analyzed so far, in which the RTN was deleted and $CO_2$/pH chemosensitivity abrogated, suffered to various

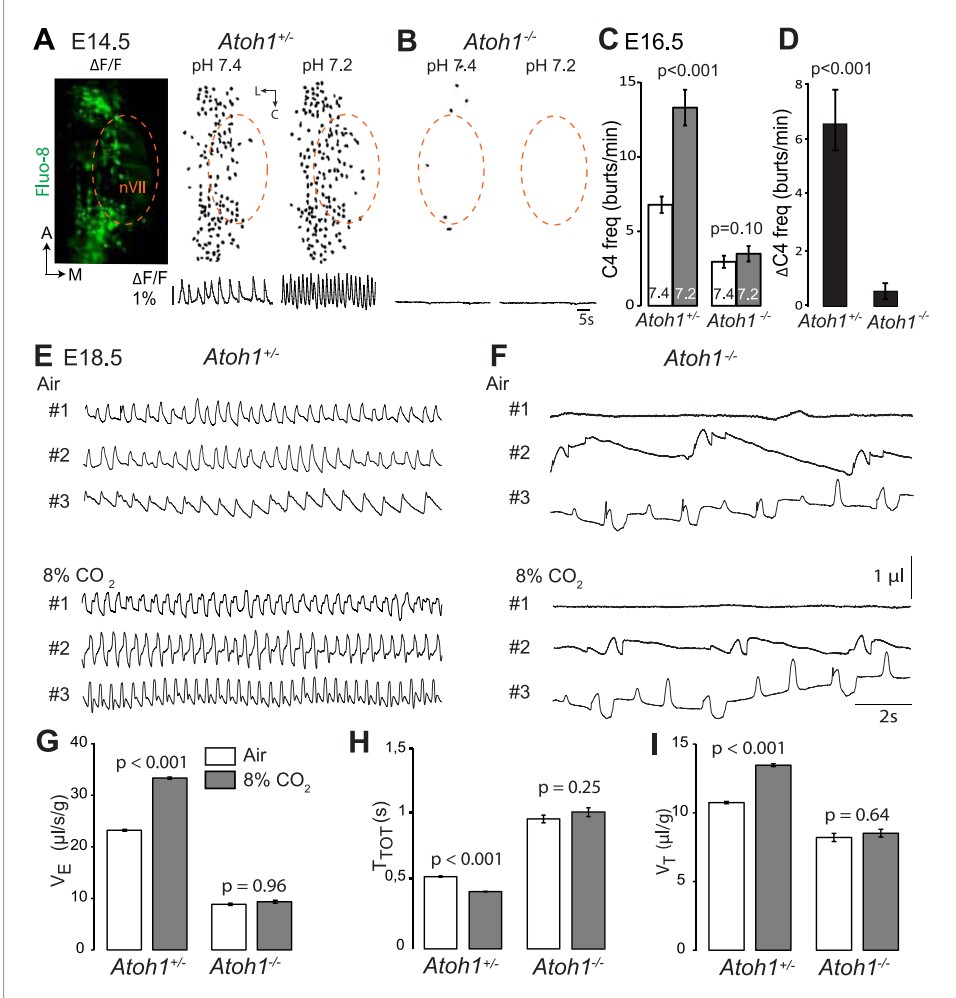

**Figure 2.** Absence of a functional RTN and lack of $CO_2$ chemosensitivity in $Atoh1^{-/-}$ ($Atoh1^{CreERT2/CreERT2}$) mice. (**A**) Left, ventral view centered on nVII (dotted outline) showing Fluo-8 AM fluorescence changes ($\Delta F/F$) of RTN cells in an E14.5 $Atoh1^{+/-}$ brainstem preparation. Right, RTN cell maps (top) and average population activity traces (bottom) at pH7.4 and pH7.2. The vertical black bar in front of the traces represents a 1% $\Delta F/F$ change as indicated. (**B**) Corresponding maps and tracings for an $Atoh1^{-/-}$ embryo attesting to the absence of a functional RTN. (**C**) Quantification of C4 burst frequencies in baseline (white bar) and low pH (gray bar) conditions for $Atoh1^{+/-}$ (n = 11) and $Atoh1^{-/-}$ (n = 8) E16.5 preparations. Baseline frequency dropped by 56% in the mutants and was not affected by acidification. (**D**) Change of C4 frequency ($\Delta$C4) induced by a pH challenge in $Atoh1^{+/-}$ and $Atoh1^{-/-}$ preparations. (**E** and **F**) Representative examples of plethysmographic recordings of three live E18.5 $Atoh1^{+/-}$ and $Atoh1^{-/-}$ pups (#1–#3) delivered by Caesarean section and breathing normal or hypercapnic air as indicated. (**G–I**) Mean values of ventilation ($V_E$) (**G**), breath duration ($T_{TOT}$) (**H**) and tidal volume ($V_T$) (**I**) of control (n = 31) and mutant (n = 8) pups breathing normal or hypercapnic air, excluding periods of apnea.

extents from a lack of specificity. In these mutants, potentially all Phox2b-expressing cells (*Dubreuil et al., 2008*) or all those derived from dB2 progenitors (*Dubreuil et al., 2009*) or from rhombomeres 3 and 5 (*Ramanantsoa et al., 2011*) could have caused the respiratory defect. Others have used local viral injection in combination with a general *Phox2b* promoter to activate or silence the RTN (*Marina et al., 2010*; *Abbott et al., 2011*; *Basting et al., 2015*). However, one drawback of this approach is that the nearby Phox2b⁺ catecholaminergic neurons, which also control breathing but whose connectivity and functions differ from that of the RTN, are transduced as well. We therefore sought to design genetic strategies to make the *Atoh1* or *Phox2b* mutations more selective.

**Table 1.** Relative changes (%) in breathing parameters in response to hypercapnia (8% $CO_2$) measured by plethysmography in E18.5 or P0 pups of the indicated genotypes

| Genotypes | $\Delta V_E$ | p-value | $\Delta T_{TOT}$ | p-value | $\Delta V_T$ | p-value | n |
|---|---|---|---|---|---|---|---|
| *control* | $60.9 \pm 8.9$ | <0.001 | $-19.5 \pm 2.8$ | <0.001 | $27.5 \pm 5.7$ | <0.001 | 51 |
| *Atoh1$^{-/-}$* | $-3.7 \pm 8.1$ | 0.53 | $6.7 \pm 10.4$ | 0.58 | $-1.3 \pm 4.9$ | 0.62 | 8 |
| *P2b::CreBAC1;Atoh1$^{lox/lox}$* | $-8.0 \pm 12.8$ | 0.35 | $14.6 \pm 10.6$ | 0.63 | $0.4 \pm 10.9$ | 0.55 | 13 |
| *Atoh1$^{Cre}$;Phox2b$^{lox/lox}$* | $18.7 \pm 13.5$ | 0.91 | $-1.9 \pm 8.1$ | 0.31 | $9.2 \pm 9.2$ | 0.45 | 10 |
| *P2b::FLPo;Atoh1$^{FRTCre}$;Vglut2$^{lox/lox}$* | $20.3 \pm 8.6$ | 0.20 | $-3.6 \pm 6.1$ | 0.31 | $16.7 \pm 9.2$ | 0.25 | 12 |

In the E15.5 brainstem, co-expression of *Phox2b* and *Atoh1* is restricted to periVII and periV neurons (*Dubreuil et al., 2009*; *Rose et al., 2009b*) (and see below). They are the only cells in the brain with this transcriptional signature, since Phox2b and Atoh1 are absent from more rostral regions (*Rose et al., 2009a*; *Hirsch et al., 2013*). To monitor these cells specifically, we constructed the *Atoh1$^{FRTCre}$* line (*Figure 4A*), which expresses Cre recombinase from the *Atoh1* locus conditionally to the action of FLP recombinase, and partnered it with the *P2b::FLPo* allele that expresses *FLP* from the *Phox2b* promoter (*Hirsch et al., 2013*). In mice harboring the *P2b::FLPo* and *Atoh1$^{FRTCre}$* alleles, Cre recombinase will thus be active selectively in cells with a history of both *Phox2b* and *Atoh1* expression (*Figure 4B*). Indeed, in triple transgenic *P2b::FLPo;Atoh1$^{FRTCre}$;Tau$^{GFPnLacZ}$* embryos, brainstem expression of the *Tau$^{GFPnLacZ}$* reporter was specific for periV and periVII cells (*Figure 4C,D*), verifying that these cells are the only ones with a history of *Phox2b* and *Atoh1* expression (hereafter *Phox2b$^{on}$/Atoh1$^{on}$* cells). Importantly, the nearby catecholaminergic neurons, despite their expression of Phox2b, lie outside the *Phox2b$^{on}$/Atoh1$^{on}$* intersectional population and are thus excluded by our approach.

In the embryo, the *Atoh1*-expressing cells in the vicinity of the facial nucleus represent a large fraction of RTN neurons as defined by Phox2b and *Vglut2* expression and the absence of catecholaminergic and motoneuronal markers (85%, *Dubreuil et al., 2009*). However, the functional characteristics of these cells have not been examined. In *P2b::FLPo;Atoh1$^{FRTCre}$;R26$^{tdTomato}$* embryos, *Phox2b$^{on}$/Atoh1$^{on}$* cells specifically express the fluorescent reporter tdTomato. The tdTomato$^+$ cells located ventral of the facial nucleus co-expressed Phox2b but not the motoneuronal marker Islet1,2 (*Figure 5A*). They increased in numbers between E14.5 and E18.5, probably because of the time required for full recombination by Cre recombinase expressed from the *Atoh1* locus, which starts only at E12.5 (*Dubreuil et al., 2009*). We then used calcium imaging to show that all tdTomato$^+$ cells in the RTN region analyzed (297/297 cells from 7 different preparations) were rhythmically active and responded to acidification by increasing their bursting frequencies (*Figure 5B–D*). Importantly, the burst frequencies and responses to acidification of the tdTomato–positive cells in *P2b::FLPo; Atoh1$^{FRTCre}$;R26$^{tdTomato}$* embryos were identical to those of the global population of oscillating RTN neurons (p = 0.5 and 0.7, respectively) (*Figure 5C*). In E14.5 hindbrain preparations, tdTomato$^+$ cells recorded in the whole-cell configuration (n = 4) featured spontaneous rhythmic burst discharges of action potentials whose frequency increased at pH 7.2 (*Figure 5E,F*). These spontaneous bursts appeared as all-or-none voltage-dependent events that could be initiated by depolarizing current pulses (data not shown) and forced to terminate prematurely by negative current pulses (*Figure 5G*). Together, these data show that the *Phox2b$^{on}$/Atoh1$^{on}$* cells share the functional signature of embryonic RTN neurons: they oscillate spontaneously and accelerate their rhythm in low pH (*Dubreuil et al., 2009*; *Thoby-Brisson et al., 2009*) and thus are part of the intrinsically rhythmic network of embryonic RTN cells characterized by *Thoby-Brisson et al. (2009)*.

We then tested the capacity of *Phox2b$^{on}$/Atoh1$^{on}$* cells to elicit C4 activity upon optogenetic stimulation as shown above for the global population of oscillating RTN neurons (see *Figure 1E*). In *P2b::FLPo;Atoh1$^{FRTCre}$;Ai32* embryos, expression of ChR2-YFP depends on Cre recombinase (Ai32 line, *Madisen et al., 2012*) and is thus specific for *Phox2b$^{on}$/Atoh1$^{on}$* cells. Single light pulses of 150 ms duration, delivered to the RTN region in E15.5 brainstem-spinal cord preparations at a fixed delay of 4 s after a spontaneous C4 burst, triggered a motor burst response in C4 and facial neurons (*Figure 5H*) in around 70% of the cases (130 out of 188 stimulations from 4 preparations). The

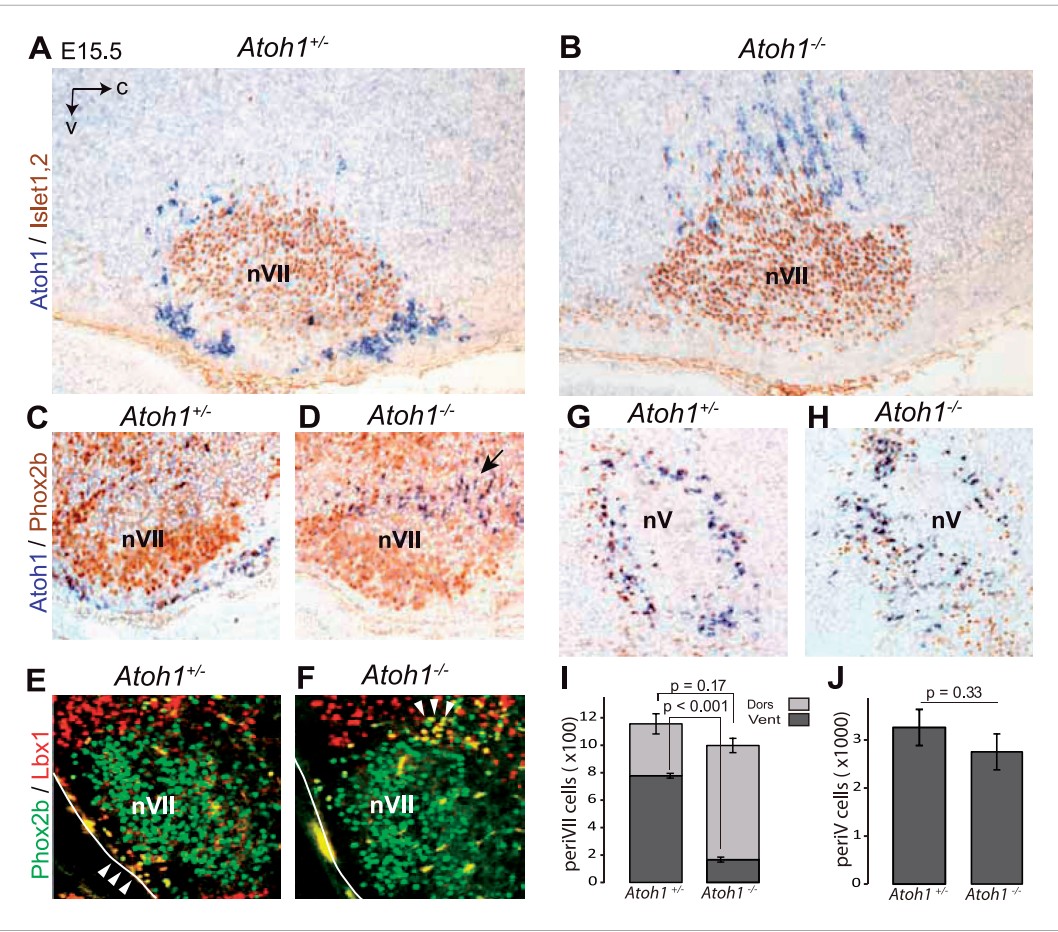

**Figure 3**. Ventral to dorsal shift of periVII and normal location of periV cells in $Atoh1^{-/-}$ embryos. (**A** and **B**) Combined in situ hybridization (ISH) with a 3′ UTR probe of $Atoh1$ (blue) and immunohistochemistry (IHC) with anti-Islet1,2 antibodies (brown) to visualize nVII on sagittal sections through the hindbrain from E15.5 $Atoh1^{+/-}$ and $Atoh1^{-/-}$ embryos. In $Atoh1^{-/-}$ embryos, the neurons identified by expression of the $Atoh1$ 3′ UTR, which would have expressed Atoh1 protein normally, are depleted ventral of nVII visualized by Islet1,2 labeling, but accumulate dorsally. (**C** and **D**) Combined ISH with a 3′ UTR probe for $Atoh1$ and IHC for Phox2b on sagittal (**C** and **D**) (anterior at left) sections through the medulla of E15.5 $Atoh1^{+/-}$ and $Atoh1^{-/-}$ embryos as indicated. The arrow points to the dorsally shifted periVII cells that express the $Atoh1$ 3′ UTR and Phox2b. The PeriVII cells that would have normally expressed the Atoh1 protein are depleted ventral of nVII in the mutants and accumulate dorsally. (**E** and **F**) Immunofluorescence for Phox2b (green) and Lbx1 (red) on coronal sections (lateral at left) through the medulla of E15.5 $Atoh1^{+/-}$ and $Atoh1^{-/-}$ embryos. The Lbx1$^+$/Phox2b$^+$ periVII cells (yellow) are depleted ventral of nVII but accumulate dorsally (arrowheads). (**G** and **H**) Combined ISH with a 3′ UTR probe for $Atoh1$ and IHC for Phox2b on coronal sections (lateral at left) through the pons of E15.5 $Atoh1^{+/-}$ and $Atoh1^{-/-}$ embryos showing the normal location of periV cells in the mutants. (**I**) Quantification of the dorsal and ventral population of periVII cells identified by $Atoh1$ 3′ UTR expression at E15.5. Their total numbers were not significantly different from the controls in $Atoh1^{-/-}$ embryos, but 83 ± 1.5% are located dorsally in the mutants vs 32 ± 4.1% in the controls. (**J**) Quantification of periV cells identified by $Atoh1$ 3′ UTR expression at E15.5.

parameters of spontaneous and evoked bursts measured on 150 spontaneous and 123 evoked C4 bursts, respectively, were not significantly different (spontaneous vs evoked: amplitude: 201 ± 152 arbitrary units (a.u.) vs 183 ± 117 a.u., p = 0.08; time to peak: 228 ± 17 ms vs 242 ± 14 ms, p = 0.53; half-width: 450 ± 21 ms vs 442 ± 17 ms, p = 0.76; decay time: 510 ± 25 ms vs 519 ± 22 ms, p = 0.79). The latency of the C4 response and its variability (666 ± 20 ms for 123 photostimulations), which cannot be easily explained by the time required for polysynaptic propagation of activity, probably reflect the delays caused by processes necessary for burst intitiation in the RTN (192 ± 12 ms, see above) and in the preBötC (*Kam et al., 2013*).

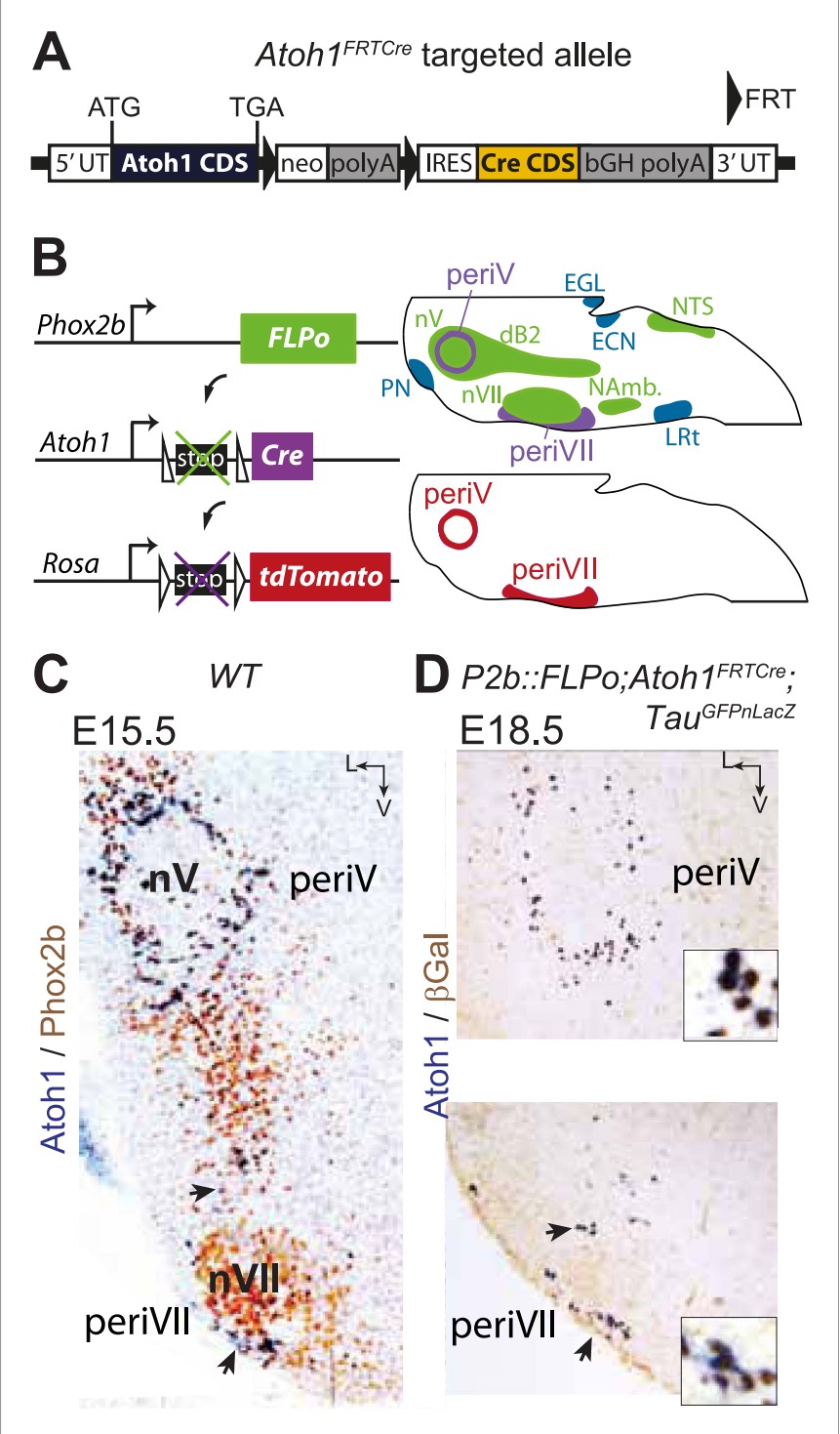

**Figure 4**. Targeting of *Phox2b^on*/*Atoh1^on* cells by intersectional genetics. (**A**) Schematic of the *Atoh1^FRTCre* allele generated by homologous recombination in ES cells. The targeted *Atoh1* locus contains from 5′ to 3′ the *Atoh1* CDS up to and including the stop codon, a neomycin resistance cassette with three consecutive SV40 polyadenylation sequences flanked by FRT sites, an IRES sequence, the Cre recombinase CDS (orange) followed by a bovine growth hormone polyadenylation sequence and the *Atoh1* 3′ UTR. FLP recombinase-mediated recombination will remove the neomycin cassette with the polyadenylation signals allowing for expression of Cre from the *Atoh1* locus. (**B**) Left, schematic of the *P2b::FLPo* (top) and *Atoh1^FRTCre* (middle) alleles and the Cre recombinase-responsive R26^tdTomato indicator allele (bottom). Right, top, sagittal brainstem cartoon schematizing the structures with a history of Phox2b

*Figure 4. continued on next page*

*Figure 4. Continued*

(green) or Atoh1 (blue) (*Rose et al., 2009a* and data not shown) expression and the intersectional population expressing or having expressed both (purple). Below, the periV and periVII cells that will express tdTomato in *P2bFLPo;Atoh1$^{FRTCre}$*;R26$^{tdTomato}$ embryos (red). The cells expressing FLPo recombinase from the Phox2b promoter will activate Cre expression from the *Atoh1* locus in the intersectional population, which in turn will activate tdTomato expression (or nuclear βgalactosidase expression if *Tau$^{GFPnLacZ}$* is used as indicator allele). EGL, external granular layer, ECN, external cuneate nucleus, LRt, lateral reticular nucleus, NAmb., nucleus ambiguus, NTS, nucleus of the solitary tract, PN, pontine nuclei. (**C** and **D**) Combined ISH for *Atoh1* and IHC for Phox2b (**C**) or nuclear βgalactosidase (**D**) on coronal sections through the hindbrain from E15.5 wild-type (WT) (**C**) or E18.5 *P2b::FLPo; Atoh1$^{FRTCre}$*;*Tau$^{GFPnLacZ}$* (**D**) embryos. The periV and periVII cells are the only cells in the brainstem co-expressing *Phox2b* and *Atoh1* (**C**) and also the only cells with a history of both *Phox2b* and *Atoh1*expression (**D**). The arrows in panels **C** and **D** point to the periVII cells. The insets in panel **D** show close ups of periV (top panel) and periVII cells (bottom panel), double-labeled for *Atoh1* and βgalactosidase.

Therefore, the *Phox2b$^{on}$/Atoh1$^{on}$* subset of RTN neurons is able to entrain phrenic nerve activity and thus inspiration as previously reported in adult rats of neurons in the RTN region defined by *Phox2b* expression alone (*Abbott et al., 2009*, *2011*). In the latter experiments, activation of breathing could not be unambiguously traced back to RTN neurons classically defined as non-catecholaminergic and non-cholinergic, since as many nearby catecholaminergic as non-catecholaminergic and some cholinergic neurons also expressed ChR2 (*Abbott et al., 2011*). These results are consistent with our data obtained in reduced preparations. Together, they show that the RTN regulates inspiration, not only expiration as postulated by others (*Janczewski and Feldman, 2006*). In fact, opinions vary regarding the role of Phox2b-expressing RTN neurons in active expiration. *Abbott et al. (2011)* found that activation of Phox2b$^+$ cells in the RTN region also triggered active expiration and *Marina et al. (2010)* that silencing these neurons in the in situ perfused brainstem-spinal cord preparation reduced hypercapnia-evoked expiratory activity. By contrast, *Pagliardini et al. (2011)* reported that photostimulation of cells expressing ChR2 from a pan-neuronal promoter in the RTN region generated mainly active expiration with little effect on inspiration. These authors attributed the effect seen in their experiments to neurons that did not express Phox2b and are thus not targeted by our approach. Similarly, *Tupal et al. (2014)* reported that in E18.5 brainstem-spinal cord preparations, a lumbar motor output and thus active expiration persisted after the depletion of *Atoh1* in RTN neurons.

A conditional genetic strategy that combines the criteria of *Phox2b* and *Atoh1* expression will target periV in addition to periVII neurons. We found that periV neurons are not essential to $CO_2$ chemosensitivity. To show this, we compared phrenic nerve activity in a standard brainstem-spinal cord preparation with one from which the pons and thus periV cells have been removed by sectioning (*Figure 6A*). At E16.5, in baseline conditions, sectioning accelerated C4 discharges by suppressing the inhibitory influence on the respiratory rhythm generator (RRG) exerted by pontine neurons (*Ito et al., 2000*; *Hilaire et al., 2004*). However, the response to a pH challenge was fully preserved (*Figure 6B,C*) (p = 0.1, n = 34). Hence, at least in vitro, periV neurons do not seem to contribute to the $CO_2$/pH response. Conversely, affecting periVII cells, but leaving periV neurons intact, suffices to eliminate the $CO_2$ response: indeed, our previous results showed that conditional inactivation of *Phox2b* or activation of the toxic *P2b$^{27Alacki}$* allele using *Egr2$^{Cre}$* as Cre driver resulted in the lack of responsiveness to $CO_2$ or acidification (*Dubreuil et al., 2009*; *Ramanantsoa et al., 2011*), despite the fact that *Egr2$^{cre}$* directs recombination in periVII (*Dubreuil et al., 2009*; *Ramanantsoa et al., 2011*) but not in periV cells (*Figure 6D–G*).

## Inactivation of *Atoh1* in *Phox2b$^{on}$* neurons abrogates $CO_2$ chemosensitivity

We restricted the *Atoh1* null mutation to *Phox2b$^{on}$/Atoh1$^{on}$* cells by partnering a floxed *Atoh1* locus with *Cre* expressed from the *Phox2b* promoter (*P2b::CreBAC1*) (*D'Autreaux et al., 2011*). Like *Phox2b*, *Cre* was switched on in the dB2 progenitor domain (*Figure 7A*) from which RTN neurons arise (*Dubreuil et al., 2009*; *Hirsch et al., 2013*). In *P2b::CreBAC1; Atoh1$^{lox/lox}$* embryos, *Atoh1* expression should be lost selectively in cells that express or have expressed *Phox2b*, that is, in *Phox2b$^{on}$/Atoh1$^{on}$*

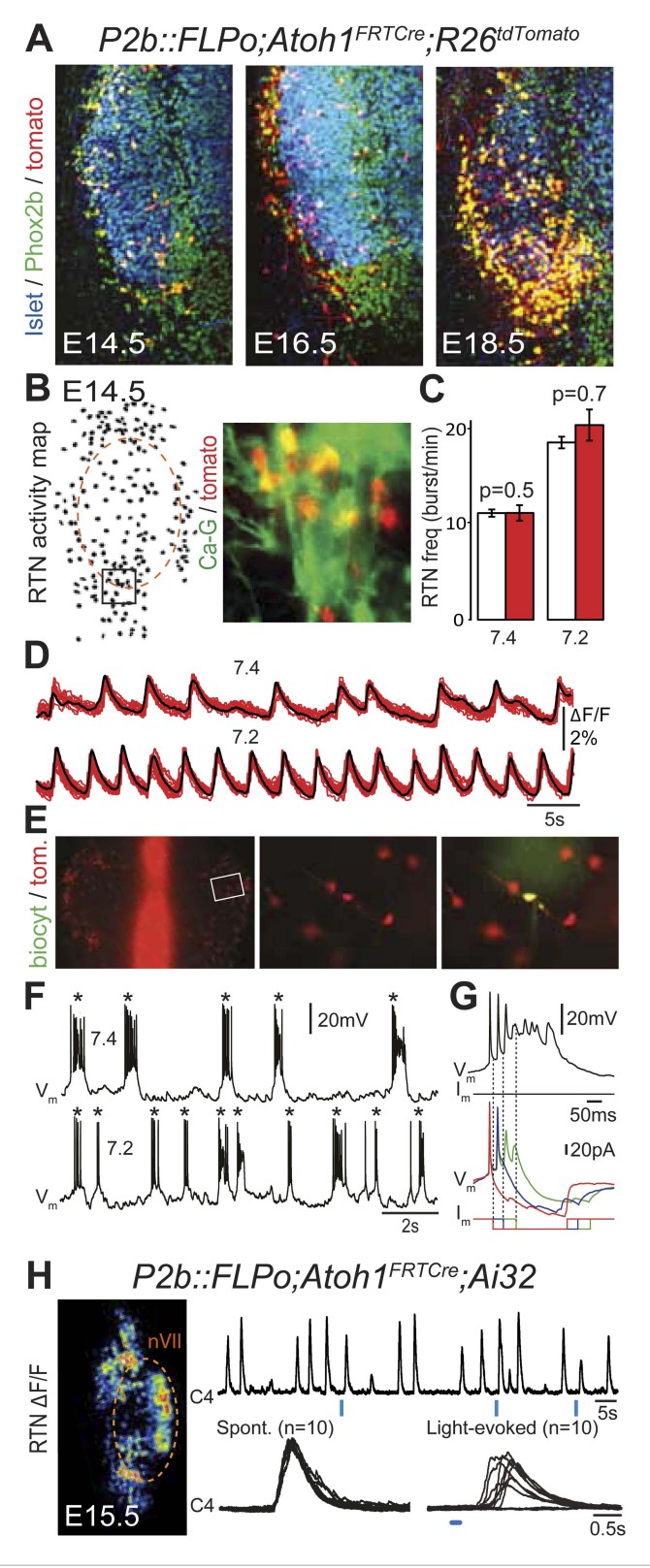

**Figure 5.** Functional characterization of *Phox2b^on^/Atoh1^on^* cells. (**A**) Ventral view of triple immunofluorescence for Islet1,2, Phox2b and tdTomato over the facial area in *Atoh1^FRTCre^;P2b::FLPo;R26^tdTomato^* embryos showing *Phox2b^on^/Atoh1^on^* cells (yellow) at E14.5 (left), E16.5 (middle) and E18.5 (right). (**B**) RTN activity map and right, close up of the

*Figure 5. Continued*

boxed area showing *Phox2b$^{on}$/Atoh1$^{on}$* cells (red) loaded with Calcium Green-1 AM (Ca-G) (yellow). (**C**) Histogram showing that the burst frequencies of the global population of oscillating RTN neurons (white bars, n = 27 preparations) are not significantly different from that of the tdTomato$^+$ (thus *Phox2b$^{on}$/Atoh1$^{on}$*) cells in *Atoh1$^{FRTCre}$;P2b::FLPo; R26$^{tdTomato}$* embryos (red bars, n = 6 preparations) at pH7.4 (7.4) and pH7.2 (7.2). (**D**) Superimposed traces showing spontaneous rhythmic Ca-G fluorescence changes of 16 individual *Phox2b$^{on}$/Atoh1$^{on}$* cells (red traces) and their average (black trace) at pH7.4 (top, 7.4) and pH7.2 (bottom, 7.2). Note the frequency increase in low pH. (**E**) Left panel, ventral view of an E14.5 brainstem preparation showing *Phox2b$^{on}$/Atoh1$^{on}$* tdTomato$^+$ RTN neurons (red) on either side of the midline (marked by the auto-fluorescence of the holding mesh) during an electrophysiology experiment. Middle panel, close up of the boxed area showing *Phox2b$^{on}$/Atoh1$^{on}$* cells (red) and right, one biocytin-filled *Phox2b$^{on}$/Atoh1$^{on}$* cell during whole-cell patch-clamp recording. (**F**) Top row, trace of the membrane potential trajectory of the biocytin-labeled cell in **E** showing spontaneous burst discharges of action potentials (indicated by *) at pH7.4 (7.4); bottom row, same trace at pH7.2 (7.2). Note the increased frequency of bursts at low pH. (**G**) Close up of a burst discharge (top) that can be curtailed by negative current pulses (bottom) applied 10 ms (red traces), 50 ms (blue traces) or 100 ms (green traces) after onset of the first action potential of the burst. (**H**) Photostimulation of *Phox2b$^{on}$/Atoh1$^{on}$* cells expressing ChR2 at E15.5 evokes C4 motor responses. Left panel, ΔF/F changes in the RTN and adjacent nVII (dotted line) following photostimulation of the RTN region. Right, at top, integrated activity of the C4 root showing spontaneous activity and bursts of activity evoked by light pulses (blue bars) triggered 4 s after a spontaneous burst. Bottom left, close up of a set of 10 superimposed spontaneous (Spont.) C4 bouts of activity and right, a corresponding set of light-evoked (Light-evoked) C4 responses, synchronized on the light pulse (blue bar), showing their variable latencies.

cells. Indeed, *Atoh1* expression was abrogated in the ventrolateral medulla at E15.5 and as early as E12.5 (*Figure 7B,C*) attesting to an efficient recombination. Predictably, *Atoh1* expression was preserved outside of the *Phox2b* lineages, that is, in the rhombic lip and the migrating cells of the anterior extramural stream. In *P2b::CreBAC1; Atoh1$^{lox/lox}$* embryos, as in *Atoh1$^{-/-}$* embryos, the normally *Atoh1*-expressing periVII cells that can be detected in these mutants by double Phox2b/Lbx1 labeling (*Pagliardini et al., 2008*; *Rose et al., 2009b*) were quantitatively preserved (p = 0.11). However, their distribution was massively shifted from ventral to dorsal (75 ± 2.6% dorsally located Phox2b$^+$/Lbx1$^+$ cells in the mutants vs 25 ± 2.3% in the controls, p < 0.001, n = 3 and 4 for controls and mutants, respectively) and the ventral *Phox2b$^{on}$/Atoh1$^{on}$* RTN neurons correspondingly depleted, in agreement with previous results (*Huang et al., 2012*) (*Figure 7D–F*).

In *P2b::CreBAC1;Atoh1$^{lox/lox}$* mutants, we looked for cells displaying the functional hallmarks of the embryonic RTN: rhythmic activity, acid-sensitivity and entrainment of the RRG (*Thoby-Brisson et al., 2009*). In E14.5 brainstem preparations, calcium activity in the RTN region of the conditional mutants was similar to that observed in *Atoh1$^{-/-}$* embryos: few active cells (8 ± 2 cells, n = 5 vs 119 ± 11 in the controls, n = 13), absence of synchronized rhythmic fluorescent changes and unresponsiveness to acidification (*Figure 8A,B*). At E16.5, the frequency of C4 discharges was reduced as in other mutants with impaired RTN function (*Dubreuil et al., 2009*; *Ramanantsoa et al., 2011*) (by an average of 49%, p < 0.001, n = 12 and 15 for mutants and controls, respectively) (*Figure 8C,D*). The mutant preparations retained a residual response to a pH challenge, which, however, was much attenuated (an increase in C4 frequency by 51 ± 10% above the pH 7.4 value vs 106 ± 17% in the controls, p < 0.001, n = 12 and 15 for mutants and controls, respectively) and remained below the baseline value of the controls (*Figure 8C,D*).

We then examined the C4 response to photostimulation of the RTN region in E15.5 brainstem-spinal cord preparations from mutants harboring the *ChR2-YFP* allele expressed from the *Vglut2* promoter. We introduced the *ChR2-YFP* allele (*Vglut2::ChR2-YFP*) (*Hägglund et al., 2010*) in a *P2b::CreBAC1;Atoh1$^{lox/lox}$* background and its *P2b::CreBAC1;Atoh1$^{lox/+}$* control. In E15.5 control preparations, stimulation of the RTN region by single light pulses of 150 ms duration, delivered at a fixed delay of 4 s after a spontaneous burst (n = 147 photostimulations in 6 preparations), systematically elicited a synchronized calcium response in the RTN neurons, followed by a motor burst in the facial nucleus and the C4 roots (*Figure 8E*). Here again, the latency of the C4 response and its variability (623 ± 16 ms for 147 photostimulations) reflect probably the engagement of processes necessary for burst intitiation in the RTN and the preBötC. A comparison of spontaneous and evoked motor burst parameters, measured on 150 spontaneous and 147 evoked C4 bursts, showed that the amplitudes and kinetics of spontaneous and evoked motor bursts were not significantly different (spontaneous vs evoked amplitude: 310 ± 11 a.

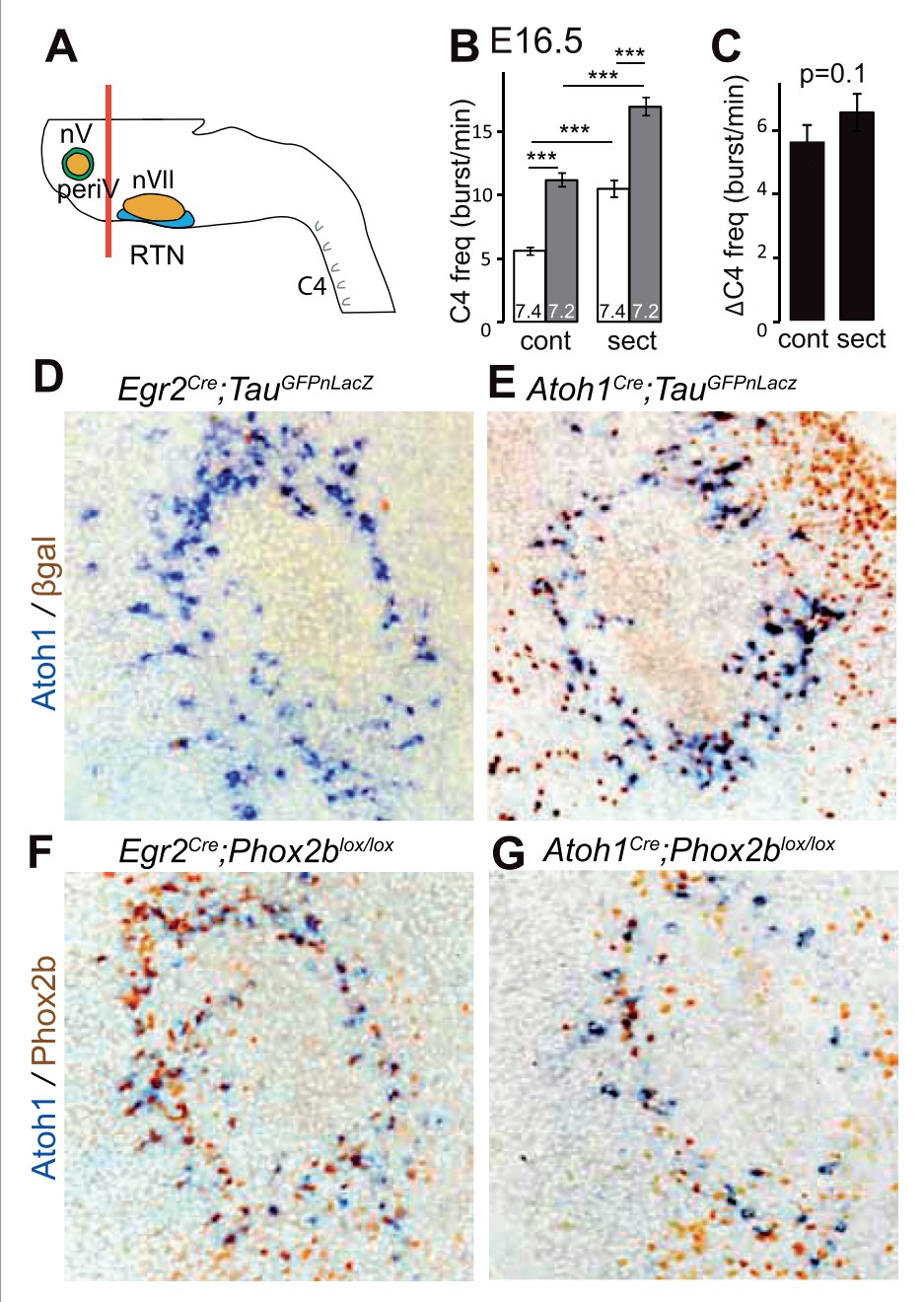

**Figure 6**. PeriV cells are not essential for the respiratory chemoreflex. (**A**) Schematic of a standard brainstem-spinal cord preparation showing the level of the section (red line) to eliminate the pons. (**B**) Comparison of C4 burst frequencies before and after sectioning in E16.5 wild-type preparations in normal and low pH conditions. The section (sect) increases baseline (white bar) frequency but preserves the response to acidification (grey bar), ***p < 0.001. (**C**) The changes of C4 frequency ($\Delta$C4) induced by the pH challenge before (cont) and after sectioning (sect) are not significantly different from each other. (**D** and **E**) Combined ISH for *Atoh1* and IHC for nuclearly localized βgalactosidase on coronal sections through the E15.5 pons, in the indicated genotypes. With *Egr2* as Cre driver, expression of the *Tau*[GFPnLacZ] reporter allele is not activated in periV cells (**D**), but it is expressed there when Cre is provided by *Atoh1*[cre] (**E**). (**F** and **G**) Combined ISH for *Atoh1* and IHC for Phox2b on coronal sections through the E15.5 pons in the indicated genotypes. In the presence of *Egr2*[Cre], Phox2b expression by the *Atoh1*[+] cells is preserved in *Phox2b*[lox/lox] embryos (**F**), but not when Cre is provided by *Atoh1*[Cre] (**G**). PeriV cells are not in the *Egr2* lineage and can thus not be affected in *Egr2*[Cre];*Phox2b*[lox/lox] or *Egr2*[Cre];*P2b*[27Alacki] mice, in which the respiratory chemoreflex is abrogated.

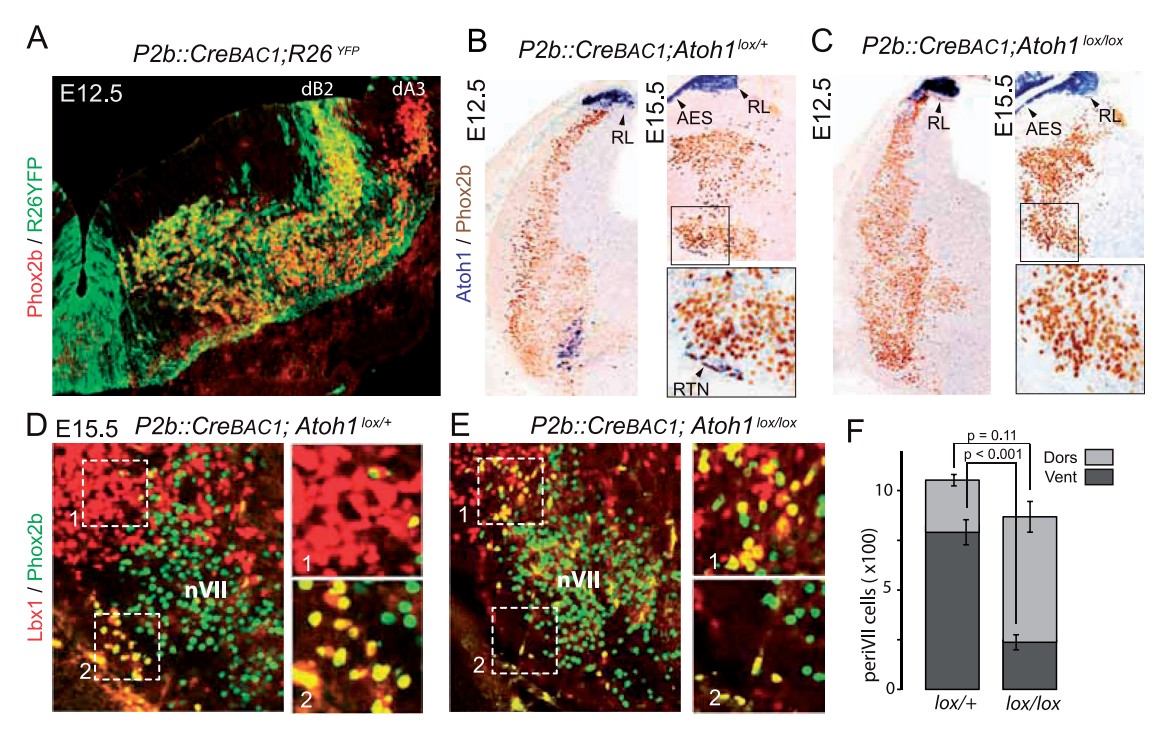

**Figure 7**. Selective removal of *Atoh1* in RTN precursors produces a ventral to dorsal shift of periVII neurons. (**A**) Combined Phox2b (red) and YFP (green) staining on a transverse section of an E12.5 *P2b::CreBAC1;R26ᵛᶠᵖ* (**Srinivas et al., 2001**) hindbrain showing efficient recombination of the reporter allele in dB2 progenitors. In the dA3 domain, where Phox2b and Cre are switched on postmitotically, newly born cells are still YFP⁻ but express it during their ventral migration. (**B** and **C**) Combined *Atoh1* ISH with a CDS probe of *Atoh1* and Phox2b IHC on coronal hindbrain sections of P2b::CreBAC1;*Atoh1ˡᵒˣ/⁺* (**B**) or *P2b::CreBAC1;Atoh1ˡᵒˣ/ˡᵒˣ* embryos (**C**) at E15.5 or E12.5 as indicated. Cre recombinase expressed from the Phox2b promoter removes *Atoh1* message in the RTN precursors but not in the rhombic lip (RL) or the cells of the anterior extramural stream (AES) of *Atoh1ˡᵒˣ/ˡᵒˣ* embryos. Higher magnifications of the boxed areas are shown below. (**D** and **E**) Immunofluorescence for Lbx1 (red) and Phox2b (green) on coronal E15.5 hindbrain sections of the indicated genotypes (dorsal at top, lateral on the left). The double-labeled periVII cells are in yellow, the insets show close ups corresponding to the boxed areas. The Lbx1⁺/Phox2b⁺ cells accumulate dorso-laterally of nVII in *P2b::CreBAC1;Atoh1ˡᵒˣ/ˡᵒˣ* mutants. (**F**) Quantification of the Lbx1⁺/Phox2b⁺ periVII cells at E15.5 located ventrally (Vent) or dorsally (Dors) of nVII. In *P2b::CreBAC1;Atoh1ˡᵒˣ/⁺* controls, 25 ± 2.3% of the periVII neurons are located dorso-laterally of nVII vs 75 ± 2.6% in *P2b::CreBac1; Atoh1ˡᵒˣ/ˡᵒˣ* mutants.

u. vs 317 ± 13 a.u., p = 0.33; time to peak: 194 ± 9 ms vs 202 ± 12 ms, p = 0.61; half-width: 446 ± 15 ms vs 454 ± 17 ms, p = 0.73; decay time: 487 ± 16 ms vs 509 ± 20 ms, p = 0.40). In the *P2b::CreBAC1; Atoh1ˡᵒˣ/ˡᵒˣ* background, the same light stimulus resulted in the activation of only a few RTN neurons and completely failed to trigger bursts in C4 and the facial nucleus (n = 317 photostimulations in 6 preparations) (*Figure 8F*). Therefore, the residual Phox2b and Lbx1 co-expressing RTN precursors that reach the medullary surface (see *Figure 7F*) are unable to mount a motor response.

Reasoning that the dorsally misrouted periVII cells may underlie the residual chemoresponse in E16.5 preparations, we compared the functional status of ventral and dorsal cells in E15.5 slice preparations (*Figure 9A*). In control preparations, the full complement of rhythmic cells revealed by their calcium activity was located ventral of the facial nucleus. In the *P2b::CreBAC1;Atoh1ˡᵒˣ/ˡᵒˣ* mutants, by contrast, rhythmic cells were massively depleted ventrally, but a population of neurons with synchronized rhythmic activity was now found dorso-laterally of the facial nucleus (n = 5) (*Figure 9B,C*). The dorsally located mutant periVII cells were thus distinct from wild-type dorsal cells and functionally resembled wild-type ventral cells. The ectopic periVII cells in the *P2b::CreBAC1; Atoh1ˡᵒˣ/ˡᵒˣ* mutants accelerated their rhythm in low pH conditions (*Figure 9C,E*). They had conserved their glutamatergic nature (*Figure 9F*) and could thus mediate the attenuated C4 response to acidification in the brainstem-spinal cord preparations. In *Atoh1⁻/⁻* embryos (n = 5), the dorso-laterally located rhythmic cells still responded to low pH but were fewer and less synchronized (*Figure 9D,E*), which may explain the complete absence of a response.

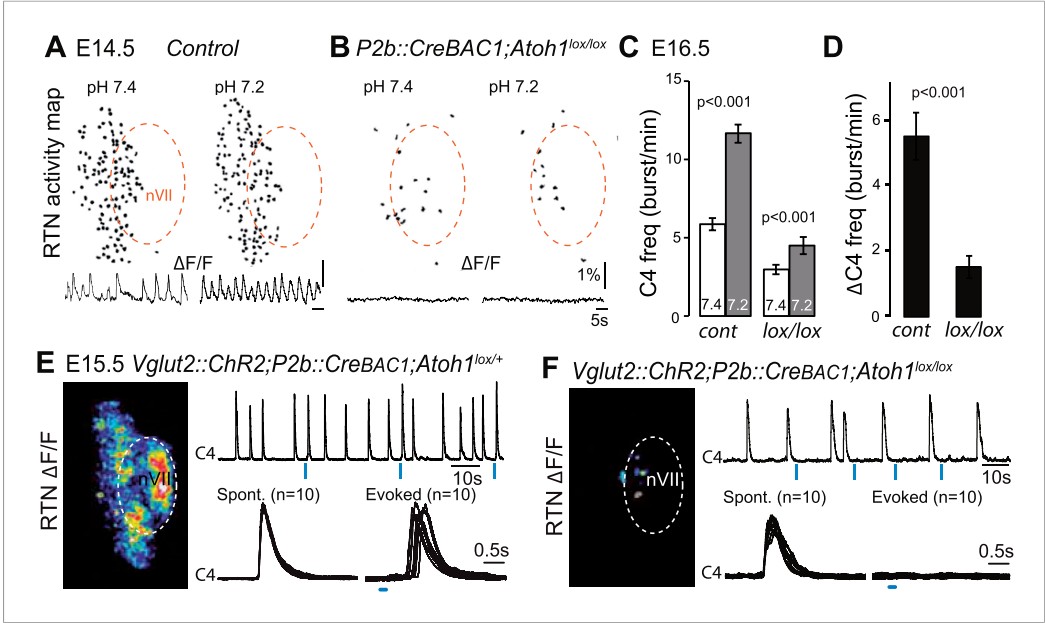

**Figure 8**. Functional consequences of selective inactivation of *Atoh1* in *Phox2b^on* cells. (**A** and **B**) RTN activity maps and average population activity traces at E14.5 in control and *P2b::CreBAC1;Atoh1^lox/lox* mutants at pH7.4 and pH7.2 showing absence of a functional RTN in the mutant. (**C**) Quantification of C4 burst frequencies in baseline (white bars) and low pH (gray bars) conditions in control and mutant E16.5 preparations. Baseline frequency dropped by 49% in the *P2b::CreBac1;Atoh1^lox/lox* mutants (lox/lox) and increased at pH 7.2 by 51 ± 10% above the pH7.4 value vs 106 ± 17% in the controls (cont). (**D**) Corresponding change of C4 frequency (ΔC4) induced by a pH challenge in *P2b::CreBAC1;Atoh1^lox/lox* (lox/lox) compared to control (cont) preparations. (**E**) Left panel, photostimulation of the RTN region at E15.5 evokes ΔF/F changes in the RTN and adjacent nVII (dotted line) in *Vglut2::ChR2;P2b::CreBAC1; Atoh1^lox/+* preparations. Right panel, above, integrated C4 activity showing spontaneous bursts and bursts evoked by light pulses (blue bars) triggered 4 s after a spontaneous burst. Bottom left, close-up of a set of 10 superimposed spontaneous (Spont.) C4 bouts of activity and right, a corresponding set of light-evoked (Evoked) C4 responses, synchronized on the light pulse (blue bar) that appeared with variable latencies after the stimulus. (**F**) In *P2b:: CreBAC1;Atoh1^lox/lox* mutants, ΔF/F responses to light are limited to a few cells in the RTN and nVII (left panel). Right panel, at top, integrated activity of C4 showing the reduced frequency of the C4 rhythm and systematic failures of photostimulation (blue bars) to evoke C4 responses; bottom, close-up of spontaneous C4 bouts of activity (Spont.) and lack of response to light (blue bar).

Finally, we examined by plethysmography the newborn and adult *P2b::CreBAC1;Atoh1^lox/lox* animals, which survived (around 60%). In newborn mutants at the day of birth (P0), mean ventilation ($V_E$) in normal air was reduced by 16%, (from 22.8 ± 0.3 to 19.2 ± 0.6 μl/s/g, p < 0.001, n = 13 and 31 for mutants and controls, respectively). Hypercapnic air increased $V_E$ of the controls by 60 ± 13% (p < 0.001) but had no effect on the mutants (p = 0.34) (*Figure 10A* and *Table 1*), and this was true also at P2.5 (*Figure 10B–D*) (p = 0.18, n = 9 and 17, for mutants and controls, respectively). Adult mutants breathing hypercapnic air responded with a 30% increase in $V_E$, corresponding to an average of 42% recovery of the $CO_2$ response with respect to the adult controls (p < 0.001).

In conclusion, *Atoh1* inactivation in *Phox2b^on* cells disrupts the RTN anatomically and functionally, severely curtails the $CO_2$/pH response in embryonic brainstem preparations and abrogates it at birth, with only a partial recovery at later stages.

## Inactivation of *Phox2b* in *Atoh1^on* neurons abrogates $CO_2$ chemosensitivity

RTN neuron development depends on *Phox2b* (*Dubreuil et al., 2009*). In a second strategy to genetically impair the RTN, we targeted the inactivation of *Phox2b* to *Atoh1^on* cells by partnering a floxed *Phox2b* locus (*Phox2b^lox*, *Coppola et al., 2010*) with *Cre* expressed constitutively from the

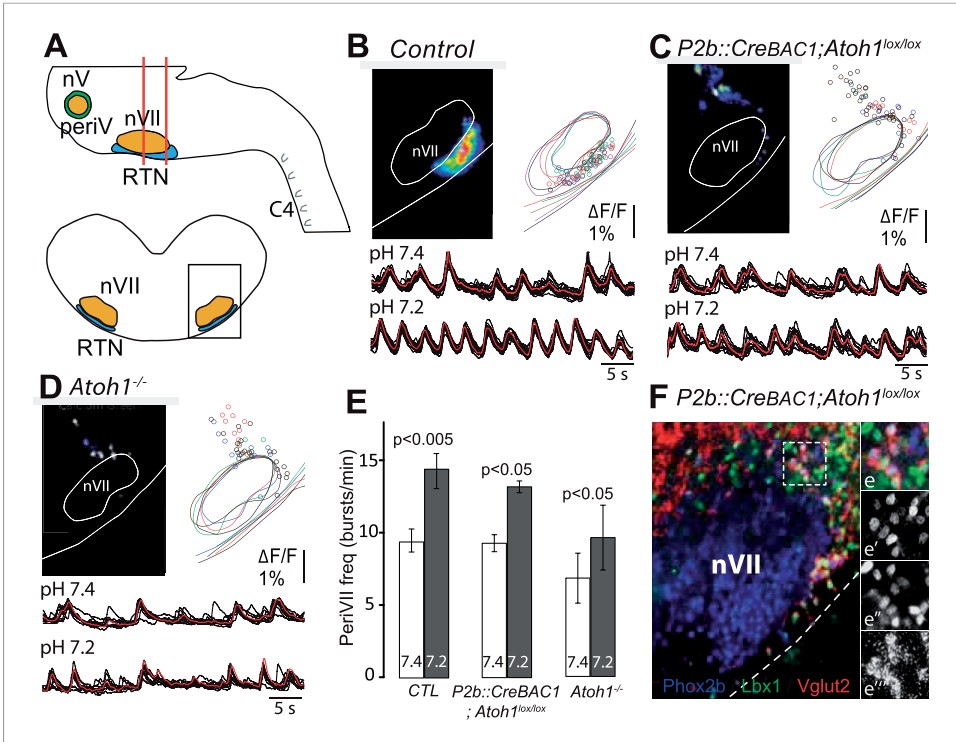

**Figure 9**. The dorsally misplaced periVII cells in *P2b::CreBAC1;Atoh1^lox/lox^* embryos are rhythmic and respond to a pH challenge. (**A**) Cartoon of the brainstem (top) showing medullary slice preparation (delimited by vertical red lines) used for calcium imaging of dorsally misplaced mutant cells (bottom, black box). (**B** and **C**) Above, ΔF/F changes and activity maps and below, superimposed traces showing spontaneous rhythmic fluorescence changes (traces corresponding to individual periVII cells in black and average trace in red) at pH7.4 (top, 7.4) and pH7.2 (bottom, 7.2) in E15.5 transverse slices from control (**B**) or *P2b::CreBAC1;Atoh1^lox/lox^* (**C**) preparations. The rhythmic cells are found ventrally in the controls and are massively displaced dorsally in the mutants. (**D**) Same experiment for *Atoh1^−/−^* preparations, showing fewer rhythmic cells that are less well synchronized. In panels **B**–**D**, the matched colors of the circles and the outlines of nVII and the medullary surface represent individual experiments (**B**, n = 9; **C**, n = 5; **D**, n = 5). (**E**) Histograms showing that the frequency of the oscillations of periVII cell is not significantly different at pH7.4 (7.4, white bars) between controls (CTL) and *P2b::CreBAC1;Atoh1^lox/lox^* or *Atoh1^−/−^* mutants and is increased by acidification to pH7.2 (7.2, gray bars) in all cases. (**F**) Combined fluorescent ISH for *Vglut2* (red) and immunofluorescent staining of Phox2b (blue) and Lbx1 (green) on coronal sections of an E15.5 *P2b::CreBAC1; Atoh1^lox/lox^* hindbrain showing *Vglut2* expression by the dorsally misplaced Phox2b+/Lbx1+ cells; (e–e′′′), close ups of the boxed area showing the overlay (e), Phox2b (e′), Lbx1 (e′′) and *Vglut2* (e′′′) expression.

*Atoh1* promoter by the *Atoh1^Cre^* allele that was generated by germ-line excision of the STOP cassette in *Atoh1^FRTCre^* by FLP recombinase (see **Figure 4A**). In embryos homozygous for the *Phox2b^lox^* allele and expressing Cre from the *Atoh1* locus, *Phox2b* will be inactivated selectively in periVII and periV cells. A caveat is that *Atoh1* and thus *Cre* are switched on in the migrating RTN precursors only at E12.5 (*Dubreuil et al., 2009*), well after Phox2b, which is already expressed in their progenitors (*Hirsch et al., 2013*). Therefore, early differentiation of RTN neurons is expected to proceed normally in *Atoh1^cre^;Phox2b^lox/lox^* embryos and functional defects to appear only during late gestation. At E15.5, most RTN neurons, identified by *Atoh1* expression and their location ventral of the facial nucleus, had already lost Phox2b, but their number had dropped by only 34 ± 6% (p = 0.013, n = 4) (*Figure 11A–C*). In brainstem-spinal cord preparations from E16.5 *Atoh1^Cre^;Phox2b^lox/lox^* mutants, the baseline frequency of C4 discharges was reduced by an average of 56% (p < 0.001), but the response to hypercapnia was preserved (p = 0.002, n = 5 and 12 for mutants and controls, respectively) (*Figure 11D,E*).

At E18.5, however, the number of *Atoh1^+^* RTN neurons in the mutants had dropped by 62 ± 2% (p = 0.001, n = 3) (*Figure 11F–H*), and the newborn *Atoh1^Cre^;Phox2b^lox/lox^* pups were completely

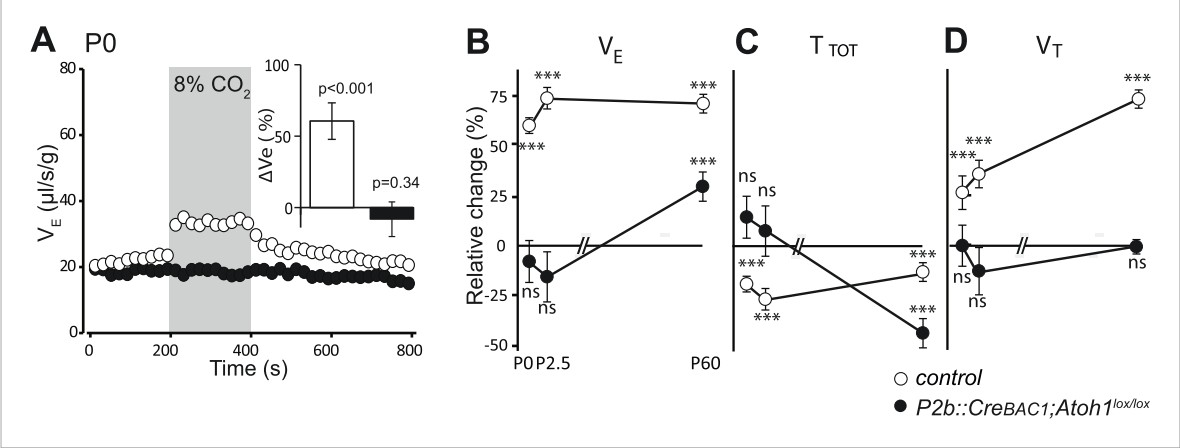

**Figure 10**. Defective CO$_2$ chemosensitivity after inactivation of *Atoh1* in *Phox2b$^{on}$* cells. (**A**) Plethysmographic recording of P0 pups. Shown are the mean values of V$_E$ in air or in response to 8% CO$_2$ in *P2b::CreBAC1;Atoh1$^{lox/lox}$* mutants (black circles) and controls (empty circles). Inset, ventilatory response to hypercapnia expressed as the percentage change relative to baseline for controls (white bar) and mutants (black bar). (**B–D**) Mean relative changes of ventilation (V$_E$) (**B**), breath duration (T$_{TOT}$) (**C**) and tidal volume (V$_T$) (**D**) measured in P0 (control n = 31; mutant N = 13), P2.5 (n = 17; N = 9) and P60 (n = 19; N = 11) mice breathing normal or hypercapnic air (8% CO$_2$). ***p < 0.001, n.s. = p > 0.05 (hypercapnic vs room air).

unresponsive to hypercapnia (p = 0.91, n = 10) while baseline ventilation was reduced by 24% (p < 0.001, from 24.4 ± 0.6 µl/s/g in the control, n = 10 to 18.4 ± 0.5 µl/s/g in the mutants, n = 10). Thus inactivating *Phox2b* in periVII and periV cells eliminates the CO$_2$ response at birth.

### Inactivation of glutamatergic synaptic transmission in Phox2b$^{on}$/Atoh1$^{on}$ neurons abrogates CO$_2$ chemosensitivity

Previous data suggested that RTN neurons are potentially glutamatergic and may thus activate the RRG by releasing glutamate (*Mulkey et al., 2004*; *Bochorishvili et al., 2012*). Proper functioning of the RRG requires glutamatergic synaptic transmission since mice with germ-line inactivation of *Vglut2*, in which the synaptic release of glutamate is blocked, die at birth from respiratory failure (*Wallen-Mackenzie et al., 2006*). To restrict *Vglut2* inactivation to *Phox2b$^{on}$/Atoh1$^{on}$* cells, we partnered the floxed *Vglut2* locus (*Vglut2$^{lox}$*) with the *P2b::FLPo;Atoh1$^{FRTCre}$* genotype (*Figure 12A*). In mice homozygous for *Vglut2$^{lox}$* and harboring at the same time the *P2b::FLPo* and *Atoh1$^{FRTCre}$* alleles, *Vglut2* inactivation will be targeted selectively to periVII and periV cells. The baseline ventilation of *P2b::FLPo;Atoh1$^{FRTCre}$; Vglut2$^{lox/lox}$* pups was only slightly reduced (by 9%; from 22.0 ± 0.7 µl/s/g in the control, n = 9, to 20.2 ± 0.6 µl/s/g in the mutants, p < 0.001, n = 12), but their hypercapnic response was abolished (p = 0.2, n = 12) (*Figure 12B* and *Table 1*). Thus the RTN mediation of the CO$_2$ response is glutamatergic and the chemoreflex abrogated by blocking glutamatergic transmission specifically in *Phox2b$^{on}$/Atoh1$^{on}$* cells.

### Discussion

In the mammalian CNS, localized networks of interneurons play critical roles in controlling behavior. It has, however, proved difficult to pinpoint the contribution of individual classes of neurons to defined behaviors. One example is the RTN. The RTN neurons have been implicated in breathing regulation by CO$_2$, but their importance for the respiratory chemoreflex in the intact brain remains controversial (*Huckstepp and Dale, 2011*; *Nattie, 2011*; *Guyenet et al., 2013*). Here we reexamined this issue using novel genetic backgrounds inspired by prior evidence that RTN neurons express and require the Phox2b and Atoh1 transcription factors (*Dubreuil et al., 2009*; *Rose et al., 2009b*). We first established that the *Phox2b$^{on}$/Atoh1$^{on}$* subset of RTN neurons are (i) rhythmically active, (ii) are activated by low pH, and (iii) both necessary and sufficient for optogenetic entrainment of the phrenic nerve. These neurons thus have the functional signatures of RTN neurons (*Abbott et al., 2009*; *Thoby-Brisson et al., 2009*; *Ramanantsoa et al., 2011*). We then show that an *Atoh1* constitutive null mutation and three conditional mutations specifically targeting *Phox2b$^{on}$/Atoh1$^{on}$* neurons and

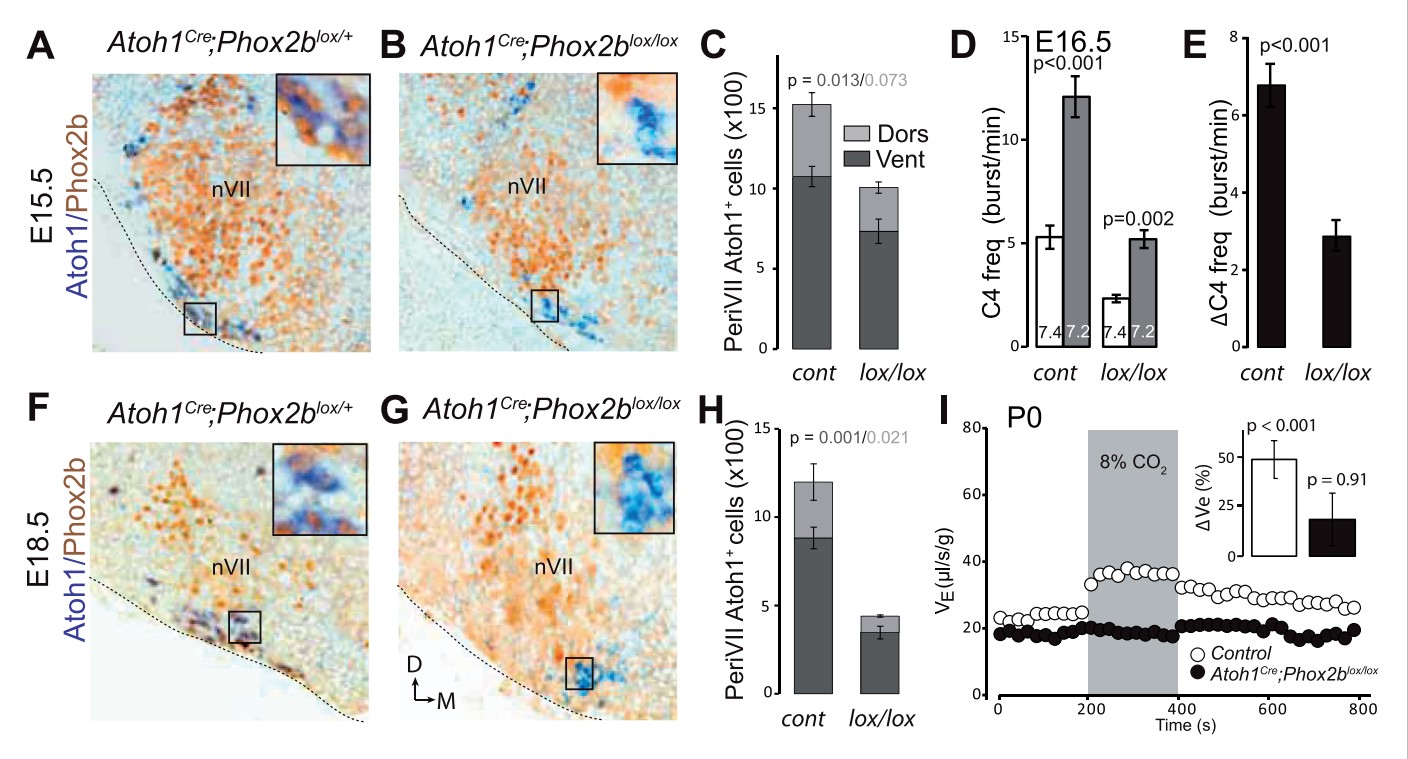

**Figure 11**. Loss of RTN neurons and the respiratory chemoreflex after selective inactivation of *Phox2b* in *Atoh1^on^* cells. (**A**, **B**, **F**, **G**) Combined ISH for *Atoh1* (blue) and IHC for Phox2b (brown) on coronal hindbrain sections from E15.5 (**A** and **B**) and E18.5 (**F** and **G**) embryos of the indicated genotypes. The insets show high magnifications corresponding to the boxed areas. (**C** and **H**) Counts of the Atoh1^+ periVII cells in control (*Atoh1^Cre^;Phox2b^lox/+^*) and mutant (*Atoh1^Cre^;Phox2b^lox/lox^*) embryos at E15.5 (**C**) and E18.5 (**H**). Total periVII cells represent $66 \pm 6\%$ of the controls at E15.5 (n = 4) and drop to $38 \pm 2\%$ at E18.5 (n = 3), while most *Atoh1^+* cells have lost Phox2b expression already at E15.5. There was no ventral to dorsal shift of periVII cells in the *Phox2b* mutants ($26 \pm 2\%$ and $21 \pm 0.3\%$ dorsally located cells in controls and mutants, respectively). (**D**) Quantification of C4 burst frequencies in baseline (white bar) and low pH (gray bar) conditions in control (cont) and *Atoh1^Cre^;Phox2b^lox/lox^* (lox/lox) E16.5 preparations. Baseline frequency dropped by 56% in the *Atoh1^Cre^;Phox2b^lox/lox^* mutants. It increased by 127% above the pH7.4 value in the mutants vs 135% in the controls. (**E**) Corresponding change of C4 frequency (ΔC4) induced by a pH challenge in *Atoh1^Cre^;Phox2b^lox/lox^* (lox/lox) compared to control (cont) preparations. (**I**) Plethysmographic recording of P0 pups. Shown are the mean values of $V_E$ in air or in response to 8% $CO_2$ in *Atoh1^Cre^;Phox2b^lox/lox^* mutants (black circles) (n = 10) and controls (empty circles) (n = 10). The inset represents the ventilatory response to hypercapnia expressed as the percentage change relative to baseline for controls (white bar) and mutants (black bar) showing the lack of a significant effect of hypercapnia.

disrupting RTN formation or function abolish the $CO_2$-evoked ventilatory response in neonates. Finally, we provide evidence that periV neurons, the only other group of neurons targeted by the conditional mutations, are not involved in the chemoreflex. We conclude that the neurons in the RTN region that have a history of both *Phox2b* and *Atoh1* expression are obligatory elements of the circuitry for breathing regulation by $CO_2$ in neonates.

While our phlethymosgraphic data in adults is in line with those of *Huang et al. (2012)*, our in vitro data on embryos contrasts with theirs and we find a lack of chemoreflex at birth. Using a distinct Phox2b::Cre line to delete Atoh1 in Phox2b lineage cells, Huang et al. reported a slowed down basal respiratory rhythm, a hallmark of RTN impairment (*Dubreuil et al., 2009*; *Ramanantsoa et al., 2011*), but a fully preserved response to acidification in brainstem-spinal cord preparations at late gestation. They did not test the chemoreflex in behaving pups, but it is only in the latter that we find a complete abrogation of the reflex. A possible explanation is that in vitro the RTN precursors misrouted to the dorsal site of the facial nucleus in the conditional *Atoh1* mutants are able to mount a motor response to bath acidification whose magnitude may depend on the precise experimental conditions. In support of this, we found that the ectopic dorsal cells in the mutants respond to low pH by accelerating their firing rate.

Our results demonstrate that the blunted $CO_2$ response at later postnatal stages in *P2b::CreBAC1; Atoh1^lox/lox^* mice is a recovery from an absent chemoreflex at birth. We found a very similar recovery of

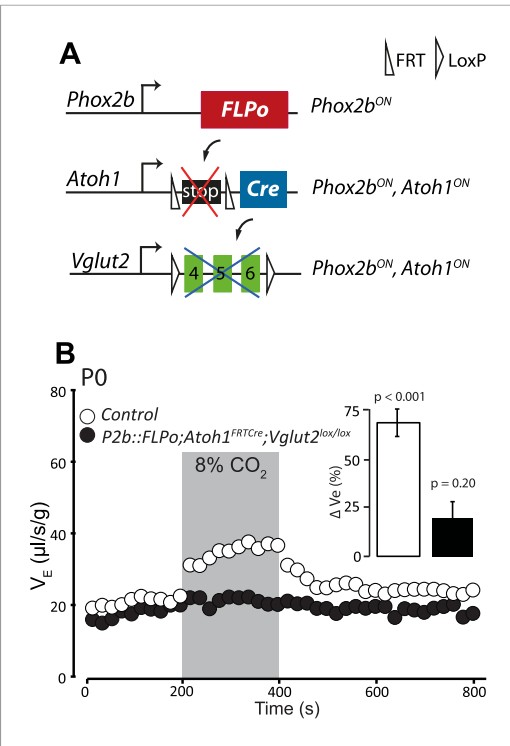

**Figure 12**. Disrupting the synaptic release of glutamate in $Atoh1^{on}/Phox2b^{on}$ cells suppresses the $CO_2$ response. (**A**) Schematic of the $P2b::FLPo$ (above) and $Atoh1^{FRTCre}$ (middle) alleles and the floxed $Vglut2$ locus (**Wallen-Mackenzie et al., 2006**) (below). The Phox2b lineage cells expressing FLPo recombinase from the Phox2b promoter will activate Cre expression from the $Atoh1$ locus selectively in $Phox2b^{on}/Atoh1^{on}$ cells. Inactivation of the floxed $Vglut2$ locus will thus be restricted to $Phox2b^{on}/Atoh1^{on}$ cells. (**B**) Plethysmographic recording of P0 $Vglut2^{lox/lox};Atoh1^{FRTCre};P2b::FLPo$ pups showing slightly slowed-down baseline ventilation in air and lack of response to 8% $CO_2$ (black circles) (n = 12) compared to controls (empty circles) (n = 9). The inset shows the ventilatory response to hypercapnia expressed as the percentage change of $V_E$ relative to baseline for mutants (black bar) and controls (white bar).

the chemoreflex in adult $Egr2^{Cre/+};Phox2b^{27Alacki/+}$ mutants (**Ramanantsoa et al., 2011**). The partial recovery of the $CO_2$ response in adults could be due to late compensation by residual RTN neurons, to a strengthening of peripheral chemoreceptor input that bypasses the RTN (**Basting et al., 2015**) or to some of the multiple sites proposed to function as $CO_2$ sensors in the brain which would now also bypass the RTN (**Nattie and Li, 2009**). The lack of the chemoreflex in many human CCHS patients (**Straus et al., 2010**; **Weese-Mayer et al., 2010**) suggests that these putative compensatory sites also have a history of Phox2b expression.

Our findings are compatible with two explanations for the loss of the chemoreflex: the Phox2b[+]/Atoh1[+] RTN neurons could be the main $CO_2$ sensors in the brain or obligatory relays funneling the chemosensory input from other cells to the RRG—or both. On the one hand, RTN neurons are exquisitely sensitive to small changes in $CO_2$/pH (**Mulkey et al., 2004**; **Onimaru et al., 2008**), also when synaptically isolated (**Wang et al., 2013**) and, in close apposition to numerous capillaries (**Onimaru et al., 2012a**), well positioned to sense $PCO_2$ in the blood. On the other, they receive afferences from many brainstem sites that contain putative chemosensors (**Rosin et al., 2006**) and respond with depolarization to activation of nearby acid-sensitive astrocytes (**Gourine et al., 2010**). In addition, RTN neurons receive excitatory connections from the carotid bodies via the nucleus of the solitary tract and increase their firing rates in response to carotid body stimulation by hypoxia (**Takakura et al., 2006**). The carotid bodies are not only the main $PO_2$ sensors, but provide also a substantial fraction of the overall $CO_2$ response (**Forster et al., 2000**; **Smith et al., 2006**). Still, we observed a complete loss of the chemoreflex in neonates with impaired RTN function. The implication is that in the perinatal period, the RTN is an obligatory relay for all forms of respiratory chemoreception whether mediated by central or peripheral chemoreceptors.

Because of its extensive connections with other chemosensory sites and the observation that its stimulation activates breathing, the RTN has been suggested to be a crucial hub for respiratory chemoreception (**Guyenet et al., 2013**). However, the results of previous loss-of-function experiments to assess the part played by RTN neurons in the chemoreflex circuitry are not entirely conclusive. In previous work, we have monitored the chemoreflex in mice in which subsets of Phox2b expressing neurons including the RTN were genetically lesioned. In these genetic backgrounds, RTN neurons were deleted and the chemoreflex lost, but the effect could not be unambiguously attributed to the RTN (**Dubreuil et al., 2008**, **2009**; **Ramanantsoa et al., 2011**). Others have impaired RTN function in behaving adult rats using pharmacological or pharmacogenetic approaches, but similar data are not available for the newborn period. Bilateral lesions of the neurokinin1 receptor-expressing neurons in the RTN region by injection of saporin conjugated to a substance P analogue have produced variable results, ranging from a modest attenuation of the chemoreflex in an earlier study (**Nattie and Li, 2002**)

to a substantial decrease in recent work (*Takakura et al., 2014*). Silencing the neurons in the RTN region by the GABA$_A$ agonist muscimol also yielded a large reduction of the hypercapnic response (*Takakura et al., 2013*). These experiments, however, lack specificity and the extent of the lesion or inhibition is difficult to control. *Marina et al. (2010)* used a pharmacogenetic approach to silence RTN neurons. Administration of allatostatin to rats which had received local injections of a lentivirus vector expressing the allatostatin receptor from an artificial *Phox2b* promoter reduced the hypercapnic response by an average of 60%. However, the approach used by Marina et al. targets the neurons in the rostral ventro-lateral medulla defined by Phox2b expression alone. They include C1 adrenergic and A5 noradrenergic neurons (*Stornetta et al., 2006*), whose contribution to the observed effect was not assessed.

RTN neurons express the vesicular glutamate transporter Vglut2 and establish synapses with preBötC neurons that resemble classic glutamatergic synapses (*Mulkey et al., 2004*; *Bochorishvili et al., 2012*), but this does not prove that they use glutamate and not other transmitters such as galanin (*Bochorishvili et al., 2012*) to excite RRG neurons. Removal of *Vglut2* and thus glutamatergic synaptic transmission selectively from $Phox2b^{on}/Atoh1^{on}$ neurons affected baseline ventilation only very slightly but eliminated the chemoreflex at birth. Therefore, glutamatergic excitatory drive by $Phox2b^{on}/Atoh1^{on}$ neurons underlies the respiratory chemoreflex while normocapnic ventilation seems to depend less on it.

A consensus about the sites and circuits that underlie the chemosensory control of breathing has yet to emerge. Our results establish that the $Phox2b^{on}/Atoh1^{on}$ RTN neurons, that is, the RTN neurons that have expressed *Atoh1* in the embryo, lie at the core of the chemoreflex circuitry. They are absolutely necessary for the ventilatory response to hypercapnia at birth and are still the major contributors in the adult. Irregular or instable respiration at birth is characteristic for preterm infants and not uncommon in babies that are born at term (*Gaultier and Gallego, 2005*). The key role of the RTN for breathing regulation at birth that we show here suggests that defective development or immaturity of the human equivalent, apart from causing CCHS (*Amiel et al., 2003*), could also underlie more common respiratory problems in the newborn period.

## Materials and methods

### Mice

The following mouse lines were used in this study:

1. $Atoh1^{FRTCre}$. To generate the $Atoh1^{FRTCRE}$ allele, we inserted a neomycin resistance cassette with three consecutive SV40 polyadenylation sequences flanked by FRT sites at the 3′ of the *Atoh1* CDS, which was followed by an IRES sequence and the Cre recombinase CDS (*Figure 4A*). The targeting vector was assembled in the K667 plasmid and electroporated into BD10(MCI-C57BL/6N Tac) ES cells. Correctly targeted ES cells were injected into C57BL/6 blastocysts. Construction of the targeting vector, ES cell manipulation and blastocyst injection were done by the Mouse Clinical Institute (Illkirch, France). The $Atoh1^{FRTCre}$ mice were genotyped for the presence of Cre by PCR (forward primer TGATGGACATGTTCAGGGATC, reverse primer GAAATCAGTGCGTTCGAACGCTAG).
2. $Atoh1^{Cre}$ expressing Cre recombinase from the *Atoh1* locus constitutively was derived from $Atoh1^{FRTCre}$ by germ-line expression of FLPe (*Rodriguez et al., 2000*). The $Atoh1^{cre}$ mice were genotyped for the presence of Cre and the absence of the *neomycin* gene (using GATCTCCTGTCATCTCACCT and ATGGGTCACGACGAGATCCT as PCR primers).
3. $Atoh1^{CreERT2}$. In the $Atoh1^{CreERT2}$ allele, the *Atoh1* CDS is replaced by the *CreERT2* sequence thus creating a null allele (*Fujiyama et al., 2009*), preserving transcription of the *Atoh1* 3′ UTR.
4. *P2b::CreBAC1*, a BAC transgenic line expressing Cre recombinase from the *Phox2b* promoter in a pattern mirroring expression of the endogenous Phox2b gene (*D'Autreaux et al., 2011*).
5. $Atoh1^{lox}$, harboring a floxed *Atoh1* locus generating a null allele upon Cre recombinase action (*Shroyer et al., 2007*). The *Atoh1* 3′ UTR is not expressed from the recombined $Atoh1^{lox}$ allele.
6. $Tau^{GFPnLacZ}$, previously termed $Tau^{mGFP}$ (*Hippenmeyer et al., 2005*) and $R26^{tdTomato}$ (Ai9, *Madisen et al., 2010*), floxed STOP reporter lines expressing GFP and nuclear βgalactosidase or tdTomato, respectively, in response to Cre recombinase action.
7. *P2b::FLPo*, a BAC transgenic line, expressing FLPo recombinase from the *Phox2b* promoter in a manner mirroring expression of the endogenous *Phox2b* gene (*Hirsch et al., 2013*).
8. *Ai32* expressing a channelrhodopsin2-YFP fusion protein driven by the *CAG* promoter in the *Rosa* locus (*Madisen et al., 2012*) in response to Cre recombinase action.
9. *Vglut2::ChR2*, a BAC transgenic line expressing a channelrhodopsin2-YFP fusion from the *Vglut2* promoter (*Hägglund et al., 2010*).

10. *Phox2b^lox* harboring a floxed Phox2b locus generating a null allele upon Cre recombinase action (*Coppola et al., 2010*).
11. *Vglut2^lox* harboring a floxed *Vglut2* locus generating a null allele upon Cre recombinase action (*Wallen-Mackenzie et al., 2006*).
12. *Egr2^Cre* expressing Cre from the *Egr2* locus (*Voiculescu et al., 2000*).

All lines have been crossed for at least four generations with C57BL/6 × DBA/2 F1 mice and were maintained on this background.

As there were no significant differences seen between mice that had only one copy of the floxed *Atoh1, Phox2b* or *Vglut2* alleles or lacked *Cre* and wild-type mice, these types of littermates were grouped together as controls, except when stated otherwise.

All animal studies were done in accordance with the guidelines issued by the European Community and have been approved by the research ethics committees in charge (Comités d'éthique pour l'expérimentation animale) and the French Ministry of Research.

## Histology and cell counting

The methods for immunofluorescence, combined bright field or fluorescent in situ hybridization (ISH) and immunohistochemistry (IHC) have been described (*Dubreuil et al., 2000*; *Hirsch et al., 2007*; *Dubreuil et al., 2008*). Riboprobes for *Atoh1* or *Vglut2* were synthesized using a DIG RNA labelling kit (Roche, Manheim, Germany) as specified by the manufacturer. *Atoh1* probes were derived either from the *Atoh1* 3′ UTR to reveal the cells that would have expressed Atoh1 protein normally in *Atoh1^CreERT2/CreERT2* embryos or from the *Atoh1* CDS in all other instances. The *Vglut2* signal was revealed by Cy5-tyramide working solution (Perkin Elmer, Waltham, MA). The primary antibodies used were: rabbit anti-Phox2b (*Pattyn et al., 1997*), chicken anti-GFP (Aves Lab, Tilgard, OR), rabbit anti-βgalactosidase (Cappel, Santa Ana, CA), guinea pig anti Lbx1 (*Muller et al., 2002*) and mouse anti-Islet1,2 (Developmental Studies Hybridoma Bank, Iowa City, IA). They were revealed for fluorescent staining by Alexa 488- (Invitrogen, Carlsbad, CA) or by Cy3- or Cy5-labelled (Jackson Immunoresearch, Suffolk, UK) secondary antibodies of the appropriate specificity, for bright field observation by biotin-labeled secondary antibodies and Vectastain ABC kit (Vector Laboratories, Peterborough, UK) revealed with 3,3′-diaminobenzamide. The pictures were captured with either a Hamamatsu ORCA-ER or a Leica DFC420C camera mounted on a Leica DM5500B microscope for observation through fluorescence or bright field optics, respectively. Brightness and contrast were adjusted in Adobe Photoshop uniformly across entire images maintaining signal linearity.

PeriVII neurons, defined as Phox2b$^+$/Atoh1$^+$ or Phox2b$^+$/Lbx1$^+$ cells surrounding the facial nucleus, were counted throughout an area starting rostrally with the first section containing facial nucleus neurons and stopping 70 μm caudal to the caudal end of the nucleus. PeriV neurons were defined as Atoh1$^+$ cells surrounding the trigeminal motor nucleus. Cell numbers represent bilateral cell counts in a 1:4 series of sections multiplied by 4. Statistical analysis of cell counts was done using a two-tailed t-test and XLSTAT software with alpha set at 0.05 and the results expressed as mean ± SEM.

## Calcium imaging and electrophysiology

The methods used for preparing brainstem–spinal cord and transverse slice preparations from embryonic day 14.5 (E14.5)–16.5 mouse embryos and maintaining them in oxygenated a-CSF have been described (*Dubreuil et al., 2009*; *Thoby-Brisson et al., 2009*; *Bouvier et al., 2010*). Briefly, brainstem-spinal cord and slice preparations were dissected in 4°C a-CSF of the following composition (in mM): 128 NaCl, 8 KCl, 1.5 CaCl$_2$, 1 MgSO$_4$, 24 NaHCO$_3$, 0.5 Na$_2$HPO$_4$, 30 glucose, pH 7.4. E15.5 transverse slices for optically recording of dorsally located periVII neurons were obtained by cutting 100 μm thick sections from caudal to rostral until a slice contained unambiguously the facial nucleus. Then a 600 μm slice was made and its caudal surface imaged.

For calcium imaging, brainstem-spinal cord or slice preparations were incubated at room temperature for 40–45 min in oxygenated a-CSF containing the cell-permeant calcium indicator dyes Calcium Green-1 AM (10 μM; Life Technologies, Paisley, UK), or Fluo-8 AM (10 μM; Teflabs, Austin, TX). Preparations were then transferred to a recording chamber (30°C) and let to recover for 30 min prior to start optical recordings using a conventional epifluorescence configuration with a FITC filter cube. Fluorescence images were captured from the slice surface and from the ventral surface of brainstem-spinal cord preparations exposing the RTN region, with a cooled Neo sCMOS camera (Andor Technology Ltd., Belfast, UK) using 10×, 20× and 40× objectives, an exposure time of 100 ms

and bin size of $4 \times 4$ for periods of 180 s using Micro-Manager software (https://www.micro-manager.org/wiki/). When calcium imaging was combined with photostimulation of *Vglut2::ChR2-YFP* preparations, the calcium imaging back-illumination was set at a minimal level using neutral density filters in the light-path, which caused no noticeable changes in the rhythmic activity of the RTN. Relative fluorescence changes ($\Delta F/F$) in response to photostimulation in *Figures 1E–G, 5H, 8E,F* show the $\Delta F/F$ image calculated by averaging a 1 s sequence acquisition (10 frames) starting at the offset of the light stimulus.

Whole-cell patch-clamp recordings were performed under visual control using differential interference contrast and infrared video microscopy, a MultiClamp 700B amplifier, a digitizing interface Digidata 1550 and the software program pClamp10 (all from Molecular Devices, Sunnyvale, CA). Patch electrodes pulled from borosilicate glass tubes (GC 150 TF; Clark Electromedical Instruments, Hamden, CT) were filled with a solution containing the following (in mM): 123 K-gluconic acid, 21 KCl, 0.5 EGTA, 3 MgCl$_2$, 10 HEPES, pH 7.2, supplemented with 1 mg/ml biocytine (Molecular Probes, Eugene, OR) and had a resistance of 4–6 M$\Omega$. Biocytin-labeled neurons were revealed with Extravidine-FITC (1:400; Sigma-Aldrich, Saint-Quentin Fallavier, France).

Phrenic nerve activity was recorded on E16.5 brainstem-spinal cord preparations using suction electrodes positioned on the fourth cervical root (C4) as described (*Thoby-Brisson et al., 2009*; *Bouvier et al., 2010*). Raw signals were amplified (High-gain AC, 7P511, Grass Technologies, Warwick, RI), filtered (bandwidth 0.1–3 kHz), integrated (time constant 50 ms, Neurolog System, Digitimer Ltd, Hertfordshire, UK) before digital sampling at 6 kHz and analysis using pClamp9 (Molecular Devices).

Values are given as mean $\pm$ SEM. Statistical significance was tested using a difference Student's *t*-test to compare data sets obtained from different mutants and a paired difference Student's *t*-test to compare the measurements obtained in two different conditions.

## Photostimulation

In optogenetic experiments, photostimulation was provided by either a 473 nm laser (Ikecool Corporation, Anaheim, CA) connected to an optical fiber placed over the RTN region on one side (*Figure 1A*), to illuminate a $250 \times 500$ μm elliptic area encompassing the RTN (laser power set at 8 mW) or by digital holography (*Lutz et al., 2008*) to target the RTN region defined by rhythmic calcium activity maps generated on-line (laser power density 1–5 mW/mm$^2$). Individual light pulses of 70 ms duration were delivered at a fixed delay of 4 s following a spontaneous C4 motor burst detected by a threshold device, or were triggered randomly in C4 silent preparations.

## Plethysmography

Breathing variables of E18.5–P60 animals were measured non-invasively in unanaesthetized, unrestrained animals using whole-body barometric plethysmography (*Chatonnet et al., 2007*). E18.5 pups delivered by Caesarean sections or P0 pups were placed under a heating lamp and were gently touched for 10 min until their breathing had stabilized. After a 7 min familiarization period in the plethysmograph chambers, breathing parameters (breath duration [$T_{TOT}$], tidal volume [$V_T$], and ventilation [$V_E$] calculated as $V_T/T_{TOT}$) were continuously monitored in apnea-free periods. E18.5 to P2.5 animals were placed in a 30 ml chamber and their ventilation was recorded continuously for a period of 800 s composed sequentially of a 200 s exposure to normal air followed by a 200 s exposure to hypercapnic air (8%CO$_2$/21%O$_2$/71%N$_2$) before normal air was resumed for the remaining 400 s. Adult animals were recorded in a larger chamber, first for 100 s in normal air, then the animals were exposed to the hypercapnic mix for 3 min before initiation of a 100 s period of recording prior to returning to normal air breathing. Calibrations were performed at the end of each recording session by injecting 2.5–5 μl of air in the chamber with a Hamilton syringe. In plots showing the continuous evolution of respiratory parameters before, during and after the hypercapnic challenge, each circle represents the mean, across all animals of a given genotype, of the parameter values binned in 20 s intervals. To quantify breathing parameter changes induced by hypercapnia, we first calculated for each animal the ratio of the mean parameter value during the last 100 s of the hypercapnic period over that during the last 100 s of the preceding normocapnic period, and then calculated the mean of these ratios in the population of control and mutant animals and expressed them as percentage ($\Delta V_E$ graphs and *Table 1*). Values are given as mean $\pm$ SEM. Statistical significance was tested using a difference Student's *t*-test to compare data sets obtained from different mutants and a paired difference Student's *t*-test to compare the measurements obtained in two different conditions.

## Image analysis

The time-series acquisitions were analyzed using a custom-made extended fork of the public domain ImageJ software with the open-source code deposited in a sourceforge.net repository (PhysImage [http://physimage.sourceforge.net/]). To determine the changes in calcium activity we first calculated a $\Delta F/F_0$ time-series using a plugin of PhysImage that implements a running moving average for baseline subtraction. This tool calculates $F_0$ by taking the fluorescence image (F) at each time-step (t) as follows: $\Delta F(t)/F_0 = (F(t) - F_0(t))/F_0(t)$, where $F_0(t) = \dfrac{\sum_{i=t}^{i+w} F(i)}{w}$ and w = 100 and the last w frames of the time-series are dropped. To identify rhythmic RTN cells and establish RTN maps of activity we performed cycle-triggered averages (CTAs) using the anatomical location of the RTN to define the region of interest (ROI) that represents broad RTN activity. Using the built-in ImageJ Z-project function, we calculated the image representing the standard deviation (SD) of the CTA and used an iterative thresholding algorithm to identify potential ROIs representing cells (*Hayes et al., 2012*). A trace representing each ROI's activity was extracted and then the ROIs were validated by calculating the power spectral density of these traces. If the peak power of each trace was not in the range 0.1–0.4 Hz, the corresponding ROI was discarded. For mutant backgrounds with no discernable RTN activity, the same strategy of detection was used except that the SD image of the full $\Delta F/F_0$ time-series was taken into account rather than SD of the image after CTA.

## Acknowledgements

We thank Véronique Dubreuil for early data on periV cells, Marie-Rose Hirsch and Zoubida Chettouh for technical assistance, Guillaume Dabée, Christophe Auger and Carmen Le Moal for expert animal husbandry and T Müller and C Birchmeier for antibodies.

## Additional information

### Competing interests

OK: Reviewing editor, *eLife*. The other authors declare that no competing interests exist.

### Funding

| Funder | Grant reference | Author |
|---|---|---|
| Agence Nationale de la Recherche | ANR-10-BLAN1410-02 | Jean-François Brunet, Gilles Fortin |
| Agence Nationale de la Recherche | ANR-32-BSV5-0011-02 | Gilles Fortin |
| Agence Nationale de la Recherche | ANR-10-LABX-54 MEMO LIFE | Jean-François Brunet |
| Agence Nationale de la Recherche | ANR11-IDEX-0001-02 | Jean-François Brunet |
| Fondation pour la Recherche Médicale | DEQ20130326472 | Gilles Fortin |
| Fondation pour la Recherche Médicale | DEQ20120323709 | Jean-François Brunet |

The funders had no role in study design, data collection and interpretation, or the decision to submit the work for publication.

### Author contributions

P-LR, MN, SA, Acquisition of data, Analysis and interpretation of data; FD'A, JAH, Conception and design, Acquisition of data, Analysis and interpretation of data; TF, MH, MH, OK, Drafting or revising the article, Contributed unpublished essential data or reagents; J-FB, Conception and design, Analysis and interpretation of data, Drafting or revising the article; GF, CG, Conception and design, Acquisition of data, Analysis and interpretation of data, Drafting or revising the article

## Ethics

Animal experimentation: The protocol for this study was approved by the National Committee on the Ethics of Animal Experiments Charles Darwin (Permit Number: Ce5/2012/065).

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
