## [Decision Letter]

Thank you for sending your work entitled “The retrotrapezoid nucleus neurons expressing *Atoh1* and *Phox2b* are essential for the respiratory response to CO_2_” for consideration at *eLife*. Your article has been favorably evaluated by Eve Marder (Senior editor) and three reviewers, one of whom is a member of our Board of Reviewing Editors.

The Reviewing editor and the other reviewers discussed their comments before we reached this decision, and the Reviewing editor has assembled the following comments to help you prepare a revised submission.

Overall the reviewers were very positive about the study (very thorough and stringent analysis; major advance in the field). There were no essential disagreements among the reviewers, so comments and minor criticisms are therefore included in their full length below.

*Reviewer 1*:

In this study, the authors examined whether cells of the retrotrapezoid nucleus (RTN) are needed for CO_2_/pH sensitivity at birth. This works contrast a previous paper from [29] proposing that these neurons are dispensable for CO_2_/pH sensitivity. By using intersectional genetic strategies and imaging of population activity in the RTN, the authors find that invalidation of *Atoh1* in *Phox2b*^+^ cells, or *Phox2b* in *Atoh1*^+^ cells or invalidation of glutamatergic transmission in *Phox2b*^+^/*Atoh1*^+^ cells changes the response of the cells to acidification in slices of the brainstem and abolishes responses to CO_2_ in newborn mice. Optogenetic activation of the cells restored and entrained respiratory rhythms indicating that *Atoh1* and *Phox2b* are required for the activation of breathing by increased CO_2_ or low pH. In summary, I do like the study. It combines genetic and imaging techniques to examine the role of RTN cells in respiration and brings in new information on the role of a subset of RTN cells in respiration. Overall a very nice piece of work.

*Reviewer 2*:

The relevance of the retrotrapezoid nucleus in the CO_2_-chemoreflex is still unresolved. Previous work by the authors and others suggested the retrotrapezoid nucleus to be critical in CO_2_ sensitivity ([10], [11]; Ramastosoa et al., 2011; Rose et al; 2009; [29]). Nevertheless, previous approaches have lacked specificity. Previous work of the authors mainly relied on the use of *Phox2b* mutant mice, and since *Phox2b* is expressed in served central and peripheral neuron types the unambiguous correlation between of physiological response, or lack thereof, and a particular neuronal class was not possible. The authors now use and elegant intersectional strategy (i.e. a combination of Cre and FlipO) to activate or to eliminate the RTN or RTN activity.

Overall I find this a very thorough and stringent analysis that in my opinion should settle the long lasting controversy. In addition, the manuscript reports new and interesting findings: Phenic nerve activity is entrained by the RTN, and the RTN is an obligatory relay station for the chemoreflex/CO_2_ sensitivity at birth and mediates other central and peripheral input. In the postnatal period, some chemoreflex/CO_2_ sensitivity is recovering in the absence of the RTN. Thus, in the mature animal additional centers participate chemoreflex/CO_2_ sensitivity, although the importance of these might be overestimated due to compensatory responses/mechanisms. A further finding is the identification of *Atoh1*^+^/*Phox2b*^+^ periV cells, and the elimination of these as candidates in the chemoreflex circuitry. The findings are of relevance for human pathology, i.e. congenital central hypoventilation syndrome, and has an impact on the understanding of more general respiratory problems in newborns.

*Reviewer 3*:

Within the reticular formation of the lower brainstem nuclei can only be defined by the congruence of multiple criteria such as location, axonal connections, transmitter phenotype, function and, ideally, genetic lineage. The retrotrapezoid nucleus (RTN) is no exception but what makes this topic of particular interest is the mounting evidence that this nucleus might be the nodal point for the homeostatic regulation of arterial PCO_2_ through breathing and that its abnormal development underpins CCHS (congenital central hypoventilating syndrome). Originally, RTN was rather vaguely identified as a collection of neurons located at the ventral surface of the brainstem. At present we know that the critical pH-activated neurons express various transcription factors such as *Atoh1*, *Egr2*, *Phox2b* etc. during development and we suspect that these neurons are glutamatergic. However, a definitive identification of the neurons responsible for the regulation of breathing by CO_2_ is still lacking and the study of these neurons is hampered by the impossibility to transduce them selectively with contemporary excitatory or inhibitory actuators (*ChR2*, etc).

In my view, the present study represents a major advance in the field because:

a) It provides both the definitive identification of the RTN neurons that are critical for the regulation of breathing by CO_2_, a question that has been of interest to physiology since the very beginning of the 20th century and;

b) It also provides the means to manipulate these neurons with complete selectivity. Using an elegant combination of intersectional genetics and integrative neurobiology (calcium imaging, whole cell recording, histology), the authors demonstrate that the critical neurons require both *Atoh1* and *Phox2b* for their correct development by describing the adverse consequences of knocking out Atoh1 or conditionally knocking out *Phox2b* in *Atoh1* expressing neurons on these neurons and the breathing response of the mice to CO_2_. The intersectional genetic strategy provides a mouse line in which the *Cre* recombinase is expressed almost exclusively by the RTN chemoreceptors. These mice will undoubtedly become the gold standard for future studies of these neurons. Using these mice, the authors show for the first time that the *Phox2bon/Atoh1on* RTN neurons use glutamate as transmitter and that these neurons and the embryonic parafacial oscillator are one and he same. The latter observation provides definitive evidence that the RTN chemoreceptors exhibit group pacemaker properties during the late developmental stages and function as an oscillator that is coupled to the preBotzinger complex.

Finally, the study shows that, at least in the perinatal period, the breathing network has essentially zero sensitivity to CO_2_ in the absence of RTN. This observation seems to put another nail in the coffin of prior theories according to which the chemoreflex is an emergent property of the entire respiratory pattern generator.

In conclusion, this is an outstanding study, worthy of being published in *eLife*.

Minor comments:

*Reviewer 1*:

The imaging part of the study should be better documented in the Materials and methods section. Too often references to published work are mentioned.

*Reviewer 2*:

1) The authors use many abbreviations for anatomical terms which, if avoided, would improve the reading of the paper.

2) Figure 4 might be confusing. Since *Cre* is only expressed after *Phox2b* recombination, the blue structures should not exist in the scheme, only the intersection of *Phox2b*/*Atoh1* that is shown in purple.

3) I do not understand Figure 5, left panel. What is delta F?

4) Figure 9 is lacking a quantification if the CO_2_/pH response.

*Reviewer 3*:

The manuscript is clear, fairly presented and very well illustrated. The following are very small points.

1) Results section, second sentence: please add the word potentially before “glutamatergic”. The present work verifies what is currently only a hypothesis.

2) Section “Designing a genetic strategy to target RTN neurons”, third paragraph. Can the authors be more specific and indicate the percentage.

3) Section “Designing a genetic strategy to target RTN neurons”, last line in fourth paragraph. The authors may already have the answer to this question (Figure 1) by examining the latency of the activation of *ChR2*-transduced RTN neurons.

4) Same section, fifth paragraph. This discussion is fine but the last statement should be somewhat modified because active expiration is actually triggered by activating *Phox2b* neurons in intact rats (1) and is suppressed by silencing these neurons in a reduced preparation, the arterially perfused juvenile rodent (37). The differences that the authors discuss are probably caused by the fact that the connections between RTN and the rest of the RPG are very complex in fully developed rodents and the effects produced by RTN activation depend on which part of the whole network is functional. For example, the effect of RTN activation on inspiration (frequency and amplitude) is very strong in conscious rats (1), mediocre in arterially perfused preparations (37) and essentially absent in some anesthetized preparations (Huckstepp et al., J of N 2015).

5) Reference 6: the TauGFPnLacz mouse is a rosa 26 line but it is not specifically mentioned in the Srinivas paper. Please indicate the source of this mouse.

6) Table 1 legend: it would be useful to restate the units for V_E_, T_TOT_ etc. and the FiCO2 level that was used in these experiments.

[Editors’ note: a previous version of this study was rejected after peer review, but the authors submitted for reconsideration. The previous decision letter after peer review is shown below.]

Thank you for choosing to send your work entitled “The retrotrapezoid nucleus neurons expressing *Atoh1* and *Phox2b* are essential for the respiratory response to CO_2_” for consideration at *eLife*. Your full submission has been evaluated by Eve Marder (Senior editor) and two peer reviewers, one of whom is a member of our Board of Reviewing Editors, and the decision was reached after discussions between the reviewers. Based on our discussions and the individual reviews below, we regret to inform you that this version of the manuscript will not be considered further for publication in *eLife*.

We have carefully reviewed your manuscript and are very enthusiastic about your novel findings. However, both reviewers feel that additional experiments are needed, to clarify that *Atoh1* and *Pho2b* neurons have indeed rhythmogenic properties and are absolutely necessary for chemosensation. The reviewers felt that this study contains some highly novel and therefore ‘provocative’ findings which due to this novelty need further experimental verification. For example, further evidence is needed to show that indeed *Atoh1* and *Pho2b* neurons are intrinsically rhythmically active and chemosensitive because this could mean that cells sensitive to acidification could also be involved in rhythm generation. Moreover, that the response of the cells to acidification is intrinsic needs further experimental proof. Finally, it should be clarified what precisely ontogenetic activation does to *Atoh1*/*Phox2b* cells. If they are as shown in the paper to generate C4 motor responses, this would mean that they are active during inspiration. In general quantitative analysis should be better presented in form of values (mean, SEM, P, N). If you are willing to perform the requested experiments, we would happily entertain a revised paper. As you know, it is *eLife*'s policy to not ask for substantial new experiments as part of a revision, and you are certainly welcome to take this manuscript to another journal at this time. That said, our interest is quite strong, so if you agree that the requested experiments would strengthen the paper, we hope you will resubmit it to *eLife*.

*Reviewer 1*:

Ruffault et al test the functional role of neurons located in the retrotrapezoid nucleus (RTN) in the control of constant CO_2_ and H^+^ concentrations in the arterial blood by genetic strategies allowing them to functionally ablate or restore the activity of a subset of RTN cells expressing the transcriptional factors *Atoh1* and *Phox2b*. They show that light-mediated stimulation of those neurons entrains the respiratory rhythm whereas eliminating the expression of both transcription factors or reducing their synaptic transmission abolishes the respiratory chemoreflex. The study uses state-of-the art imaging and electrophysiological techniques and addresses an important question. I have only few questions and comments:

1) The reasons explained in the Results for the development of new transgenic mouse lines are insufficiently explained. Please provide more concrete reasons (which *Atoh1*-dependent structures are here referred to).

2) The link between results related to Ca2^+^ imaging as well as optophysiological stimulation is weak. Provide in the Results section the information on how cells were loaded, which Ca2^+^ indicators were used and how the cells were stimulated by light (frequency, pulse duration). Moreover, I miss mean {plus minus} SEM values and P values throughout the main Results text to get a better idea on the overall quantitative changes. For example statement such as ‘triggered a more robust response’ need a value to understand what is meant by ‘more robust’. This criticism applies as already mentioned to the entire Results section. Since I cannot name all occasions were values, including the number of animals tested, should be included, I ask the authors to perform a detailed revision on this aspect.

*Reviewer 2*:

This is an interesting and provocative study by Ruffault et al. It will certainly stir the field of respiration as it is claims that the RTN neurons expressing *Atoh1* and *Phox2b* are “absolutely” necessary for chemosensation in neonates. Even though it has previously been postulated that *Atoh1* and *Phox2b* neurons are important for chemosensation, this is a new finding, because this complete and very specific knockout has never been done before. Nor has it ever been postulated that these neurons are obligatory for chemosensation. Previous studies attributed approximately 60% of the chemosensory response to these neurons. This left open the possibility that for example serotonin neurons are also involved in the response.

Despite my enthusiasm for this paper there are several open questions that need to be addressed.

1) The authors claim that most of the *Atoh1*/*Phox2b* neurons are intrinsically rhythmic and respond to acidification. The data shown are not convincing. To show they are intrinsically active needs additional experiments. The neurons need to be synaptically isolated, and ideally it should be demonstrated that the intrinsic discharge can be reset or somehow be altered by current injections. The trace shown is “absolutely” not enough. Indeed this is a major statement, since it would mean that the same neurons that are chemosensitive are also likely involved in rhythm generation. The authors state “most”; what is exactly meant with this? Please quantify.

The question of intrinsic rhythmicity is a very important conclusion, it is novel and it needs to be demonstrated very clearly, and unambiguously. Also Paul Gray and coworkers showed in a recent paper (in *eLife*) that *Atoh1* neurons are important in coordinating the generation of the rhythm. A functional discussion of how these authors envision the role of these neurons in the context of the Gray study would be important.

2) Along the same lines, the response to acidification and to demonstrate that this is intrinsic is actually a big deal. Some researchers go as far as “demanding” that such experiments are done with isolated cells to exclude indirect input from glia cells. Of course isolating cells comes with its own set of caveats. But, irrespective how this will be done, it is important to demonstrate an intrinsic chemosensitivity, and it is important to assess how sensitive these cells really are. Extreme acidification will likely affect all cells… 8% CO_2_ is very high. This is OK and actually very good to show the absence of a chemoresponse, but then one would also like to know how sensitive these neurons are.

3) The authors claim that optogenetic activation of the *Atoh1*/*Phox2b* neurons initiates a C4 burst. This is also new and indeed surprising. RTN/PFRG neurons have been implicated in the control of active expiration. If these neurons activate a C4 motor response it means that they likely activate inspiration. Did I miss something? Clearly, the figure shown is not enough, and I can't tell from the figure what optogenetic activation really does. The traces need to be expanded, and the motor response needs to be quantitatively characterized. This point is too important to be left as a qualitative mention with a figure that is too small.

---

## [Author Response]

Reviewer 1:

*The imaging part of the study should be better documented in the Materials and methods section. Too often references to published work are mentioned*.

We now better document the imaging part of the Materials and methods section:

“Calcium imaging and electrophysiology”. The methods used for preparing brainstem–spinal cord and transverse slice preparations from embryonic day 14.5 (E14.5)-16.5 mouse embryos and maintaining them in oxygenated artificial cerebrospinal fluid (a-CSF) have been described ([11]; Thoby-Brisson, 2009; [5]). Briefly, brainstem-spinal cord and slice preparations were dissected in 4°C a-CSF of the following composition (in mM): 128 NaCl, 8 KCl, 1.5 CaCl_2_, 1 MgSO_4_, 24 NaHCO_3_, 0.5 Na_2_HPO_4_, 30 glucose, pH 7.4. E15.5 transverse slices for optically recording of dorsally located periVII neurons were obtained by cutting 100 µm thick sections from caudal to rostral until a slice contained unambiguously the facial nucleus. Then a 600 µm slice was made and its caudal surface imaged. For calcium imaging, brainstem spinal cord or slice preparations were incubated at room temperature for 40-45 minutes in oxygenated a-CSF containing the cell-permeant calcium indicator dye Calcium Green-1 AM (10µM; Life Technologies), or Fluo-8 AM (10µM; Teflabs).

Preparations were then transferred to a recording chamber (30°C) and let to recover for 30 minutes prior to start optical recordings using a conventional epifluorescence configuration with a FITC filter cube. Fluorescence images were captured from the slice surface and from the ventral surface of brainstem-spinal cord preparations exposing the RTN region, with a cooled Neo sCMOS camera (Andor Technology Ltd.) using 10X, 20X and 40X objectives, an exposure time of 100 ms and bin size of 4x4 for periods of 180s using Micro-Manager software (https://www.micro-manager.org/wiki/). When calcium imaging was combined with photostimulation of *Vglut2::ChR2-YFP* preparations, the calcium imaging back-illumination was set at a minimal level using neutral density filters in the light-path, which caused no noticeable changes in the rhythmic activity of the RTN. Relative fluorescence changes (ΔF/F) in response to photostimulations in Figures 1, 5 and 8 show the ΔF/F image calculated by averaging a 1 second sequence acquisition (10 frames) starting at the offset of the light stimulus. “

Reviewer 2:

*1) The authors use many abbreviations for anatomical terms which, if avoided, would improve the reading of the paper*.

We now avoided the abbreviations nV, nVII and CB, changed into trigeminal motor nucleus, facial nucleus and carotid bodies, respectively. By contrast, we do not deem it appropriate to use the full term “retrotrapezoid nucleus” at each occurrence. RTN is an abbreviation widely used in the literature. PeriV and periVII are novel abbreviations chosen by us to designate the *Phox2b*^+^/*Atoh1*^+^ cells surrounding the trigeminal or facial nucleus, respectively and it would be akward to replace them each time by this paraphrase.

We agree. However, as our intention was to illustrate the intersectional strategy, we still believe the patterns of past or present expression of *Atoh1*^+^ and *Phox2b*^+^ should be visible in addition to the intersectional *Phox2b*^+^/*Atoh1*^+^ population on the schematic representation of the brainstem. As suggested we have re-colored the intersection in purple to correspond to P2b::FLPo; *Atoh1*^*FRTCre*^ allele-mediated recombination and re-phrased the legend to explicitly indicate that the blue structures that have expressed *Atoh1* but not *Phox2b* are based on our own data and those of Rose et al., 2009.

*2)*
Figure 4
*might be confusing. Since* Cre *is only expressed after* Phox2b *recombination, the blue structures should not exist in the scheme, only the intersection of* Phox2b/Atoh1 *that is shown in purple*.

We agree. However, as our intention was to illustrate the intersectional strategy, we still believe the patterns of past or present expression of *Atoh1*^+^ and *Phox2b*^+^ should be visible in addition to the intersectional *Phox2b*^+^/*Atoh1*^+^ population on the schematic representation of the brainstem. As suggested we have re-colored the intersection in purple to correspond to P2b::FLPo; *Atoh1*^FRTCre^ allele-mediated recombination and re-phrased the legend to explicitly indicate that the blue structures that have expressed *Atoh1* but not *Phox2b* are based on our own data and those of Rose et al., 2009.

*3) I do not understand*
Figure 5*, left panel. What is delta F*?

We now describe explicitly the ΔF/F panels in Figures 1, 5 and 8 in the Materials and methods section, in response to the comments of Reviewer 1 who asked for further details about the imaging methods.

In addition, we have added the following sentence:

“Relative fluorescence changes (ΔF/F) in response to photostimulations in Figures 1, 5 and 8 show the ΔF/F image calculated by averaging a 1s sequence acquisition (10 frames) starting at the offset of the light stimulus.”

*4)*
Figure 9
*is lacking a quantification if the CO*_*2*_*/pH response*.

We now provide a quantification of the CO_2-_induced changes of the RTN frequency for panels B-D in a novel panel E of Figure 9, the legend of which has also been changed.

Reviewer 3:

*The manuscript is clear, fairly presented and very well illustrated. The following are very small points*.

*1) Results section, second sentence: please add the word potentially before “glutamatergic”. The present work verifies what is currently only a hypothesis*.

This has been done.

*2) Section “Designing a genetic strategy to target RTN neurons”, third paragraph. Can the authors be more specific and indicate the percentage*.

This has been done (85%, taken from [11]).

*3) Section “Designing a genetic strategy to target RTN neurons”, last line in fourth paragraph. The authors may already have the answer to this question (*Figure 1*) by examining the latency of the activation of* ChR2*-transduced RTN neurons*.

We thank you for this useful comment. We have now calculated this latency and reported the value:

“A latency of 192 ± 12 ms (n=51 stimulations in three cells) was measured from the time of onset of the light stimulus to that of the first action potential of the burst response suggesting the engagement of a still unknown, slow obligatory process for burst initiation in the RTN.”

This is followed by the statement:

“The latency of the C4 response and its variability (666 ± 20 ms for 123 photostimulations), which cannot be easily explained by the time required for polysynaptic propagation of activity, probably reflect the delays caused by processes necessary for burst intitiation in the RTN (192±12 ms, see above) and in the preBötC (33).”

*4) Same section, fifth paragraph. This discussion is fine but the last statement should be somewhat modified because active expiration is actually triggered by activating* Phox2b *neurons in intact rats (*[1]*) and is suppressed by silencing these neurons in a reduced preparation, the arterially perfused juvenile rodent (*[37]*)*.

We modified this discussion. The paragraph now reads:

“Therefore, the *Phox2b*^***on***^*/Atoh1*^***on***^ subset of RTN neurons is able to entrain phrenic nerve activity […] a lumbar motor output and thus active expiration persisted after the depletion of *Atoh1* in RTN neurons.”

*5) Reference 6: the TauGFPnLacz mouse is a rosa 26 line but it is not specifically mentioned in the Srinivas paper. Please indicate the source of this mouse*.

We thank the reviewer for pointing out this error. The reference to Srinivas et al. has been replaced with the correct one (26).

*6)*
Table 1
*legend: it would be useful to restate the units for V*_*E*_*, T*_*TOT*_
*etc. and the FiCO2 level that was used in these experiments*.

Table 1 presents relative changes of breathing parameters expressed in percentage, hence no units. The FiCO2 is now mentioned in the table caption.

*[Editors’ note: a previous version of this study was rejected after peer review, but the authors submitted for reconsideration. The previous decision letter after peer review is shown below*.*]*

Reviewer 1:

*Ruffault et al test the functional role of neurons located in the retrotrapezoid nucleus (RTN) in the control of constant CO*_*2*_
*and H*^*+*^
*concentrations in the arterial blood by genetic strategies allowing them to functionally ablate or restore the activity of a subset of RTN cells expressing the transcriptional factors* Atoh1 *and* Phox2b*. They show that light-mediated stimulation of those neurons entrains the respiratory rhythm whereas eliminating the expression of both transcription factors or reducing their synaptic transmission abolishes the respiratory chemoreflex. The study uses state-of-the art imaging and electrophysiological techniques and addresses an important question. I have only few questions and comments*:

*1) The reasons explained in the Results for the development of new transgenic mouse lines are insufficiently explained. Please provide more concrete reasons (which* Atoh1*-dependent structures are here referred to)*.

We added a short description of *Atoh1* expression in the hindbrain that should clarify which *Atoh1*-dependent structures are referred to, and expanded the paragraph to explain better the reasons for the development of the new mouse lines. The new version now reads as follows:

“In the brainstem, *Atoh1* is expressed in the rhombic lip progenitors […] We therefore sought to design genetic strategies to make the *Atoh1* or *Phox2b* mutations more selective.”

We added at the end of the next paragraph: “Importantly, the nearby catecholaminergic neurons, despite their expression of *Phox2b*, lie outside the *Phox2b*^*on*^*/Atoh1*^*on*^ intersectional population and are thus excluded by our approach”.

*2) The link between results related to Ca2*^*+*^
*imaging as well as optophysiological stimulation is weak. Provide in the Results section the information on how cells were loaded, which Ca2*^*+*^
*indicators were used and how the cells were stimulated by light (frequency, pulse duration)*.

We added to the Results section the conditions for calcium imaging and optogenetic stimulation:

“In these and subsequent experiments, we used calcium imaging to monitor the rhythmic activity and response to acidification of embryonic brainstem preparations incubated in artificial cerebrospinal fluid containing either Calcium Green-1AM or Fluo-8 AM as calcium indicators”.

“At embryonic-day 14.5 (E14.5), single light pulses (473nm, 70ms, 1-5mW/mm^2^) applied to the RTN region”.

“When the light pulses were set to activate the RTN in a rhythmic manner at about twice the frequency of the ongoing endogenous rhythm”.

“Single light pulses of 150 ms duration, delivered to the RTN region in E15.5 brainstem-spinal cord preparations at a fixed delay of 4 sec after a spontaneous C4 burst”.

“… stimulation of the RTN region by single light pulses of 150 ms duration delivered at a fixed delay of 4 sec after a spontaneous burst (n=147 photostimulations in 6 preparations)”.

*Moreover, I miss mean {plus minus} SEM values and P values throughout the main Results text to get a better idea on the overall quantitative changes*.

SEM, p values and the number of animals have been added to the main text where appropriate.

*For example statement such as ‘triggered a more robust response’ need a value to understand what is meant by ‘more robust’. This criticism applies as already mentioned to the entire Results section. Since I cannot name all occasions were values, including the number of animals tested, should be included, I ask the authors to perform a detailed revision on this aspect*.

We did not find the expression “more robust” in the text.

Reviewer 2:

*This is an interesting and provocative study by Ruffault et al. It will certainly stir the field of respiration as it is claims that the RTN neurons expressing Atoh1 and Phox2b are “absolutely” necessary for chemosensation in neonates. Even though it has previously been postulated that Atoh1 and Phox2b neurons are important for chemosensation, this is a new finding, because this complete and very specific knockout has never been done before. Nor has it ever been postulated that these neurons are obligatory for chemosensation. Previous studies attributed approximately 60% of the chemosensory response to these neurons. This left open the possibility that for example serotonin neurons are also involved in the response*.

*Despite my enthusiasm for this paper there are several open questions that need to be addressed*.

The studies referred to by the reviewer (37; 29) are on adults. We agree with them since also in our hands around 60% of the chemoresponse in adult mice is contributed by the *Atoh1*^+^/*Phox2b*^+^ subset of RTN neurons. However, we focus on the critical newborn period where the chemoresponse is completely abolished when these cells are lesioned.

*1) The authors claim that most of the* Atoh1/Phox2b *neurons are intrinsically rhythmic and respond to acidification. The data shown are not convincing. To show they are intrinsically active needs additional experiments. The neurons need to be synaptically isolated, and ideally it should be demonstrated that the intrinsic discharge can be reset or somehow be altered by current injections. The trace shown is “absolutely” not enough*.

We now include whole-cell patch-clamp recordings on the tdTomato+ cells of *P2b::FLPo;Atoh1*^*FRTCre*^*;R26*^*tdTomato*^ embryos, i.e. on *Phox2b*^on^/*Atoh1*^on^ cells, and demonstrate that the spontaneous bursts are altered by current injections, as requested by the reviewer (panels E and F of the new Figure 5). This shows that the *Phox2b*^+^/*Atoh1*^+^ neurons behave like the embryonic parafacial oscillator described by Thoby-Brisson et al. , i.e. the global population of embryonic RTN neurons defined by their oscillating properties (64): the rhythmic burst discharges of both populations appear as all-or-none voltage-dependent events and spontaneous bursts are curtailed by short negative current pulses applied during the burst (panel G of Figure 5). In addition, we already showed in panel C of Figure 5 (formerly panel F of Figure 3) that the global population of oscillating RTN neurons and the tdTomato+ cells in *P2b::FLPo;Atoh1*^*FRTCre*^*;R26*^*tdTomato*^ embryos have identical burst frequencies and pH responses. We thus feel justified in concluding that the *Phox2b*^on^/*Atoh1*^on^ cells are indistinguishable from the intrinsically active network of embryonic RTN neurons characterized electrophysiologically and pharmacologically by [64]. These results are now described in the text as follows:

“We then used calcium imaging to show that all tdTomato^**+**^ cells in the RTN region analyzed […] and thus are part of the intrinsically rhythmic network of embryonic RTN cells characterized by [64].”

*Indeed this is a major statement, since it would mean that the same neurons that are chemosensitive are also likely involved in rhythm generation. The authors state “most”; what is exactly meant with this? Please quantify*.

*The question of intrinsic rhythmicity is a very important conclusion, it is novel and it needs to be demonstrated very clearly, and unambiguously. Also Paul Gray and coworkers showed in a recent paper (in* eLife*) that Atoh1 neurons are important in coordinating the generation of the rhythm. A functional discussion of how these authors envision the role of these neurons in the context of the Gray study would be important*.

We have quantified this and found that 100% of the tdTomato+ cells (297/297 cells from 7 different preparations) were rhythmic and respond to acidification, as now stated in the text.

Also Paul Gray and coworkers showed in a recent paper (in *eLife*) that *Atoh1* neurons are important in coordinating the generation of the rhythm. A functional discussion of how these authors envision the role of these neurons in the context of the Gray study would be important.

Tupal et al. (who are not concerned with chemosensitivity) find that the *Phox2b*^+^ RTN neurons which are the focus of our paper are not important for the temporal delay between cervical and lumbar motor outputs (the topic of their paper), so that the *Atoh1*+ cells incriminated in active expiration are distinct from *Phox2b*^+^ RTN neurons. We now cite this evidence starting: “Similarly, Tupal et al (65) reported that in E18.5 brainstem-spinal cord preparations, a lumbar motor output and thus active expiration persisted after the depletion of *Atoh1* in *Phox2b* expressing neurons, giving further weight to the notion that at least in reduced preparations, active expiration depends on *Phox2b* -negative cells”.

*2) Along the same lines, the response to acidification and to demonstrate that this is intrinsic is actually a big deal. Some researchers go as far as “demanding” that such experiments are done with isolated cells to exclude indirect input from glia cells. Of course isolating cells comes with its own set of caveats. But, irrespective how this will be done, it is important to demonstrate an intrinsic chemosensitivity, and it is important to assess how sensitive these cells really are*.

Concerning the role of RTN in CO_2_ chemosensitivity one should distinguish two questions: (i) the intrinsic chemosensitivity of the RTN and (ii) its overall contribution to the chemoreflex (through its intrinsic sensitivity and/or as a relay of the CO_2_ response of other neurons). Our paper is concerned with the second question. In the Introduction we changed the sentence “The number, location and cellular nature of the central chemosensors are still controversial” into: “The nature of the central chemosensors and of the circuits that mediate the ventilatory response to CO_2_ is still controversial”, to dispel the impression that our aim is to provide evidence for the sites and types of cells that detect changes in PCO_2_.

Regarding the issue of intrinsic chemosensitivity, Guyenet and co-workers have already demonstrated that acutely dissociated *Phox2b*^+^ RTN neurons are intrinsically chemosensitive (68). Since 79% of the *Phox2b*^+^, Vglut2+ isolated cells did respond to low pH, they necessarily include the *Atoh1*^on^/*Phox2b*^on^ neurons studied in our paper, which represent over 70% of the RTN neurons defined by co-expression of *Phox2b* and *Vglut2* (11). However, the extent to which the intrinsic chemosensitivity of RTN neurons, or indeed of serotonergic neurons, also demonstrated in culture (Corcoran et al., 2009), contributes to the chemoreflex is unknown and not our focus. What our paper shows is that the *Phox2b*^on^ / *Atoh1*^on^ RTN neurons are an obligatory relay to transmit any response of the chemosensors (RTN neurons themselves, 5HT neurons or other cells) to the respiratory rhythm generator in the newborn period and around 60% (this paper and [29]) in the adult.

Our interpretation of the results is clearly stated at the end of the Abstract (”Thus, the RTN neurons expressing *Phox2b* and *Atoh1*are a necessary component of the chemoreflex circuitry”), at the end of the Introduction (“Together our findings imply that the RTN neurons that express or have expressed *Phox2b* and *Atoh1* are essential for the activation of breathing by increased CO_2_ or low pH, and that other contributors to the chemoreflex must act via the RTN or in partnership with it”) and at the end of the first paragraph of the Discussion ( “We conclude that the neurons in the RTN region that have a history of both *Phox2b* and *Atoh1* expression are obligatory elements of the circuitry for breathing regulation by CO_2_ in neonates”). Furthermore, we state in the Discussion: “Our data are compatible with two explanations for the loss of the chemoreflex: the *Phox2b*^+^/*Atoh1*^+^ RTN neurons could be the main CO_2_ sensors in the brain or obligatory relays funneling the chemosensory input from other cells to the RRG—or both”.

*Extreme acidification will likely affect all cells… 8%* CO_2_
*is very high. This is OK and actually very good to show the absence of a chemoresponse, but then one would also like to know how sensitive these neurons are*.

We explored the pH sensitivity of the oscillating RTN neurons in E14.5 brainstem preparations. See Figure 13.

Author response image 1.**DOI:**
http://dx.doi.org/10.7554/eLife.07051.018

Left, average relationship between the burst discharge rate of the RTN, measured by calcium imaging at E14.5, and extracellular pH (n=12 preparations). Right, example F/F changes of the RTN region at different pH. On average RTN neurons increase their firing by one burst every 0.04 pH unit.

These data illustrate the pH sensitivity of the RTN neuronal population over a one unit pH change and show that the RTN neurons become silent above pH 7.6 and increasingly active below this value.

However, we do not wish to include these results in the paper since doing so would make the reader think that we are dealing here with the intrinsic chemosensitivity of the cells.

*3) The authors claim that optogenetic activation of the* Atoh1/Phox2b *neurons initiates a C4 burst. This is also new and indeed surprising. RTN/PFRG neurons have been implicated in the control of active expiration. If these neurons activate a C4 motor response it means that they likely activate inspiration*.

Activation of phrenic nerve motor output and thus of inspiration by optogenetic stimulation of *Phox2b*^+^ neurons in the RTN region has been shown before in adult rats (Figure 3 in [2] and Figure 1 in [1]). The drawback of the prior results is that the effect cannot be unambiguously traced back to RTN neurons since nearby catecholaminergic and some cholinergic neurons are also activated. Regarding the control of active expiration, Pagliardini et al. (2011) use non-selective optogenetic activation of neurons in the parafacial region to elicit increases in expiratory muscle activity with little effect on inspiration. However, these authors attribute the effects to the activation of neurons that do not express *Phox2b* and thus do not belong to the subset of RTN neurons targeted in our study, and similar results have been reported by Tupal et al. The paragraph concerning this issue now reads as follows:

“Therefore, the *Phox2b*^***on***^*/Atoh1*
^***on***^ subset of RTN neurons is able to entrain phrenic nerve activity and thus inspiration as previously reported in adult rats of neurons in the RTN region defined by *Phox2b* expression alone (2; 1). […] Tupal et al (65) reported that in E18.5 brainstem-spinal cord preparations, a lumbar motor output and thus active expiration persisted after the depletion of *Atoh1* in *Phox2b* expressing neurons, giving further weight to the notion that at least in reduced preparations, active expiration depends on *Phox2b*-negative cells.”

*Did I miss something? Clearly, the figure shown is not enough, and I can't tell from the figure what optogenetic activation really does. The traces need to be expanded, and the motor response needs to be quantitatively characterized*.

We have added whole -cell patch-clamp recordings of the spontaneous and light-evoked bursts of RTN neurons from *Vglut2::Chr2-YFP* embryos to show the cellular response to photostimulation (panel D of the new Figure 1). This Figure regroups the new results with those showing the requirement for an intact pre-BötC. The results are described in a new paragraph entitled “Photoactivation of the embryonic RTN entrains the respiratory motor output”:

“We first verified the ability of embryonic RTN neurons to entrain the respiratory-like motor output using channelrhodopsin -based optogenetics. The *Phox2b*^+^ RTN neurons are glutamatergic (4) and express the glutamate transporter *Vglut2* already at embryonic stages (11). We could thus use expression of the channelrhodopsin-2-YFP (ChR2-YFP) fusion protein driven by the *Vglut2* promoter in *Vglut2::ChR2-YFP* mice (23) to stimulate embryonic RTN neurons by light. […]When the light pulses were set to activate the RTN in a rhythmic manner at about twice the frequency of the ongoing endogenous rhythm, the C4 motor bursts could be entrained to the stimuli and followed the light-imposed rhythm (Figure 1). C4 motor bursts could not be evoked when the preBötC excitability was depressed by the µ -opiate agonist D-Ala ^2^ - *N*-Me- Phe^4^- Glycol ^5^-enkephalin (DAMGO, 0.3µM, n=4 preparations) (38) (Figure 1) or its development impaired genetically in *Dbx1* null mutants (5) (n=5 preparations) (Figure 1) indicating that the motor outputs require an intact preBötC. These data suggest that malfunction of the RTN will result in lack of entrainment of the preBötC and thus of the motor output and in a slowed-down C4 activity.”

We have redrawn the expanded traces of spontaneous and evoked C4 motor bursts in panel H of Figure 5 (formerly panel G of Figure 3) and in panels E and F of Figure 8 (formerly panels H and K of Figure 6) and have added a quantitative characterization of the spontaneous and evoked motor bursts for the results shown in Figure 5 and in Figure 8.